# A large multiethnic GWAS meta-analysis of cataract identifies new risk loci and sex-specific effects

Hélène Choquet [1✉], Ronald B. Melles[2], Deepti Anand [3], Jie Yin[1], Gabriel Cuellar-Partida[4], Wei Wang [4], 23andMe Research Team[4], Thomas J. Hoffmann [5,6], K. Saidas Nair [7], Pirro G. Hysi [8,9,10], Salil A. Lachke [3,11,12] & Eric Jorgenson [1,12]

Cataract is the leading cause of blindness among the elderly worldwide and cataract surgery is one of the most common operations performed in the United States. As the genetic etiology of cataract formation remains unclear, we conducted a multiethnic genome-wide association meta-analysis, combining results from the GERA and UK Biobank cohorts, and tested for replication in the 23andMe research cohort. We report 54 genome-wide significant loci, 37 of which were novel. Sex-stratified analyses identified *CASP7* as an additional novel locus specific to women. We show that genes within or near 80% of the cataract-associated loci are significantly expressed and/or enriched-expressed in the mouse lens across various spatiotemporal stages as per iSyTE analysis. Furthermore, iSyTE shows 32 candidate genes in the associated loci have altered gene expression in 9 different gene perturbation mouse models of lens defects/cataract, suggesting their relevance to lens biology. Our work provides further insight into the complex genetic architecture of cataract susceptibility.

[1] Kaiser Permanente Northern California (KPNC), Division of Research, Oakland, CA, USA. [2] KPNC, Department of Ophthalmology, Redwood City, CA, USA. [3] Department of Biological Sciences, University of Delaware, Newark, DE, USA. [4] 23andMe Inc, Sunnyvale, CA, USA. [5] Institute for Human Genetics, UCSF, San Francisco, CA, USA. [6] Department of Epidemiology and Biostatistics, UCSF, San Francisco, CA, USA. [7] Departments of Ophthalmology and Anatomy, School of Medicine, UCSF, San Francisco, CA, USA. [8] King's College London, Section of Ophthalmology, School of Life Course Sciences, London, UK. [9] King's College London, Department of Twin Research and Genetic Epidemiology, London, UK. [10] University College London, Great Ormond Street Hospital Institute of Child Health, London, UK. [11] Center for Bioinformatics and Computational Biology, University of Delaware, Newark, DE, USA. [12] These authors jointly supervised this work: Salil A. Lachke, Eric Jorgenson. A list of 23andMe Research members and their affiliations appears in the Supplementary Information. ✉email: Helene.Choquet@kp.org

Cataracts are caused by opacification of the crystalline lens, which leads to progressive loss of vision. They can present as a developmental disorder in younger patients (congenital or pediatric cataracts) but, more commonly, occur as a disease of aging[1,2], and are a leading cause of visual impairment. Cataract formation and cataract surgery are more common in women[3]. Twin and family aggregation studies strongly support an important role for genetic factors in cataract susceptibility with heritability estimates ranging from 35 to 58%[4–9]. A recent study[10] investigating the genetic basis of eye disease reported 20 genetic loci associated with cataract at a genome-wide level of significance in the UK Biobank European sample, although none of these loci was independently replicated. It is also unclear what proportion of clinical variability these loci help explain, as well as to what contribution they have in populations of diverse ethnic background.

In this work, we present the largest and most ethnically diverse genetic study of cataract susceptibility conducted to date to our knowledge. Following a stepwise analytical approach, we conduct a genome-wide association analyses, followed by meta-analysis, including 585,243 individuals (67,844 cases and 517,399 cataract-free controls) from two cohorts: the Genetic Epidemiology Research in Adult Health and Aging (GERA)[11] and the UK Biobank (UKB)[12,13]. We test the top independently associated SNPs ($P < 5.0 \times 10^{-8}$) at each locus in 3,234,455 participants (347,209 self-reported cataract cases and 2,887,246 controls) from the 23andMe research cohort. Cohorts summary details are presented in Supplementary Data 1. We subsequently fine-map these associations[14] and examine changes in the expression of candidate genes in associated loci in 9 gene perturbation mouse models of lens defects[15,16]. We then undertook conditional, ethnic-, and sex-specific association analyses (Supplementary Fig. 1). Finally, we assess the genetic correlation between cataract and other disorders and complex traits[17].

## Results

**GWAS of cataract and meta-analysis.** We first undertook a GWAS analysis of cataract in the GERA and UKB cohorts, stratified by ethnic group, followed by a meta-analysis across all analytical strata. In the multiethnic meta-analysis, we identified 54 loci ($P < 5.0 \times 10^{-8}$; $\lambda = 1.139$ and $\lambda_{1000} = 1.0012$, which is reasonable for a sample of this size under the assumption of polygenic inheritance[18–20]), of which 37 were novel (i.e., not previously reported to be associated with cataract at a genome-wide level of significance) (Table 1, Fig. 1, and Supplementary Fig. 2). The effect estimates of 54 lead SNPs were consistent across the 2 studies (Table 1 and Supplementary Fig. 3). In 23andMe research cohort, 45 out of 51 lead SNPs available (88.2%) replicated with a consistent direction of effect at a Bonferroni corrected significance threshold of $9.8 \times 10^{-4}$ ($P$-value = 0.05/51) and additional 2 SNPs were nominally significant ($P < 0.05$) (Table 1 and Supplementary Fig. 4).

**Replication of previous cataract GWAS results.** We also investigated in GERA the lead SNPs within 20 loci associated with cataract at a genome-wide significance level in a previous study[10]. Three of the 19 available SNPs that passed QC replicated at a genome-wide level of significance in our GERA multiethnic meta-analysis or GERA non-Hispanic white sample (including *SOX2-OT* rs9842371, 5′ *LOC338694* rs79721202, and *SLC24A3* rs4814857) (Supplementary Data 2). Further, 6 additional SNPs replicated at Bonferroni significance ($P < 0.05/19 = 0.00263$), and 6 showed nominal evidence of association.

**Ethnic-specific and conditional analyses.** To determine whether there were additional signals in individual ancestry groups that did not reach genome-wide significance in the meta-analysis, we conducted ethnic-specific meta-analyses of each ancestry group. We identified three additional novel loci in the European ancestry (GERA non-Hispanic whites + UKB Europeans) meta-analysis: *EPHA4*, *CD83-JARID2*, and near *EXOC3L2* (Supplementary Fig. 5a and Supplementary Data 3). Regional association plots of the association signals are presented in Supplementary Fig. 6. To identify independent signals within the 44 genomic regions identified in the European-specific meta-analysis (Supplementary Data 4), we performed a multi-SNP-based conditional & joint association analysis (COJO)[21], which revealed 5 additional independent SNPs within 4 of the identified genomic regions, including at known loci (*CDKN2B*, *RIC8A*, and *LOC338694*) and at newly identified *DNMBP* locus (Supplementary Data 5). Neither the meta-analysis of East Asian groups nor the meta-analysis combining the GERA African American and UKB African British groups resulted in the identification of additional novel genome-wide significant findings (Supplementary Fig. 5b and 5c).

**Sex-specific analyses.** Next, we conducted genetic association analyses for interaction between genetic factors and sex, in sex-specific meta-analyses combining data from GERA and UKB. We identified two additional novel loci, *CASP7* and *GSTM2*, in the women-specific meta-analysis (GERA + UKB) (Fig. 2 and Supplementary Data 6). *CASP7* rs12777332 and *GSTM2* rs3819350 were significantly associated with cataract in women (*CASP7* rs12777332: OR = 1.06, $P = 3.41 \times 10^{-8}$; *GSTM2* rs3819350: OR = 1.06, $P = 2.10 \times 10^{-8}$) but not in men (*CASP7* rs12777332: OR = 1.01, $P = 0.25$; *GSTM2* rs3819350: OR = 1.01, $P = 0.25$) (Supplementary Fig. 7). While we confirmed the women-specific association at the *CASP7* locus in the 23andMe replication cohort, the sex-specific association at the *GSTM2* was not validated (Supplementary Data 6). Further, among the loci identified in the multiethnic meta-analysis (GERA + UKB), we observed significant differences in the effect sizes and significance of association at five loci: one locus, *DNMBP-CPN1*, was strongly associated with cataract in women but not in men (*DNMBP-CPN1* rs1986500, OR = 0.94, $P = 5.04 \times 10^{-11}$ in women, and OR = 1.01, $P = 0.40$ in men; $Z = -5.03$, $P = 2.44 \times 10^{-7}$) (Supplementary Fig. 8 and Supplementary Data 6); and four loci, *QKI*, *SEMA4D*, *RBFOX1*, and *JAG1*, were strongly associated in men than women (*QKI* 6:163840336, OR = 0.94, $P = 1.23 \times 10^{-10}$ in men, and OR = 0.99, $P = 0.21$ in women; $Z = -3.95$, $P = 3.89 \times 10^{-5}$; *SEMA4D* rs62547232, OR = 1.15, $P = 1.83 \times 10^{-9}$ in men, and OR = 0.98, $P = 0.33$ in women; $Z = 5.03$, $P = 2.43 \times 10^{-7}$; *RBFOX1* rs7184522, OR = 1.07, $P = 9.10 \times 10^{-12}$ in men, and OR = 1.03, $P = 0.0020$ in women; $Z = 2.98$, $P = 1.43 \times 10^{-3}$; *JAG1* rs3790163, OR = 0.92, $P = 3.14 \times 10^{-12}$ in men, and OR = 0.96, $P = 9.63 \times 10^{-4}$ in women; $Z = -2.95$, $P = 1.59 \times 10^{-3}$) (Fig. 2 and Supplementary Data 7 and Supplementary Fig. 8). Similarly, we observed significant sex differences in the effect sizes and significance of association in the 23andMe replication cohort for the following loci: *SEMA4D*, *RBFOX1*, and *JAG1*. Regional association plots illustrate the sex-specific association signals (Supplementary Fig. 7).

**Variants prioritization.** We adopted a Bayesian approach (CAVIARBF)[14] to compute variants likelihood to explain the observed association at each locus and derived the smallest set of variants that has a 95% probability to include the causal origin of the signals (95% credible set). Nine sets included a single variant (Supplementary Data 8) such as rs62621812 (*ZNF800*), rs1014607 (*BAMBI-LOC100507605*), rs1428885924 (*NEK4*), rs1679013 (*CDKN2B-DMRTA1*), rs1539508 (*LOC100132354*), rs73238577 (*RFC1-KLB*), rs17172647 (*IGFBP3-TNS3*), rs73530148 (*ALDOA*),

**Table 1 Cataract loci identified in the combined (GERA + UKB) GWAS multiethnic meta-analysis and replication in 23andMe research cohort.**

| SNP | Chr | Pos | Locus | Alleles A1/A2 | Combined Meta-Analysis | | Replication in 23andMe | | Direction of Effect (GERA-UKB-23andMe) |
|---|---|---|---|---|---|---|---|---|---|
| | | | | | OR (SE) | P | OR (SE) | P | |
| rs2073017 | 1 | 10822808 | CASZ1 | C/T | 0.95 (0.0081) | 1.21E-10 | 0.98 (0.0042) | 1.40E-06 | --- |
| **rs3176459** | 1 | 51437247 | *CDKN2C* | G/A | 0.95 (0.007) | 1.63E-12 | 0.98 (0.0036) | 7.71E-10 | --- |
| rs71646944 | 1 | 82151352 | *ADGRL2* | T/C | 1.05 (0.0087) | 1.26E-09 | 1.04 (0.0045) | 5.60E-18 | +++ |
| rs10633030 | 1 | 118160005 | *FAM46C* | CT/C | 1.05 (0.0081) | 1.48E-08 | 1.03 (0.0041) | 2.56E-11 | +++ |
| **rs4745** | 1 | 155106227 | *DPM3-KRTCAP2* | T/A | 0.96 (0.0066) | 3.26E-09 | 1.01 (0.0207) | 0.66 | --+ |
| rs2982459 | 1 | 169028716 | *LINC00970* | G/A | 0.96 (0.0071) | 7.76E-09 | 0.97 (0.0037) | 1.58E-12 | --- |
| **rs12593** | 1 | 227172290 | *ADCK3* | T/C | 1.06 (0.0067) | 2.41E-16 | 1.04 (0.0035) | 2.16E-33 | +++ |
| rs890069 | 2 | 12867504 | Near *TRIB2* | T/C | 0.96 (0.0069) | 1.03E-08 | 0.98 (0.0035) | 1.45E-8 | --- |
| rs10210444 | 2 | 28859979 | *PLB1* | A/G | 1.05 (0.0084) | 1.09E-09 | 1.04 (0.0043) | 1.49E-19 | +++ |
| rs57604689 | 2 | 143577407 | *LRP1B-KYNU* | C/T | 0.95 (0.0082) | 6.74E-11 | 0.97 (0.0043) | 5.88E-14 | --- |
| rs140625707 | 2 | 218529842 | *DIRC3* | C/CT | 1.08 (0.0132) | 1.35E-09 | 1.06 (0.007) | 2.52E-18 | +++ |
| rs62237590 | 3 | 25437648 | *RARB* | C/A | 1.05 (0.0086) | 5.37E-09 | 1.02 (0.0047) | 4.61E-07 | +++ |
| rs35173917 | 3 | 52760984 | *NT5DC2* | C/CT | 0.94 (0.0086) | 5.49E-13 | NA | NA | -NA |
| rs35256080 | 3 | 63823014 | *ATXN7* | AT/A | 1.05 (0.0069) | 8.25E-12 | 1.02 (0.0035) | 3.05E-08 | +++ |
| **rs10663094** | 3 | 181363464 | *SOX2-OT* | ACT/A | 1.08 (0.0068) | 7.30E-31 | 1.05 (0.0038) | 2.22E-32 | +++ |
| rs73238577 | 4 | 39403760 | Near *KLB* | C/T | 1.08 (0.0137) | 4.52E-09 | 1.02 (0.0076) | 0.01486 | +++ |
| rs72868578 | 4 | 81951344 | *C4orf22-BMP3* | A/C | 1.12 (0.017) | 4.44E-11 | 1.07 (0.0091) | 3.08E-12 | +++ |
| rs1539508 | 6 | 43868986 | *LOC100132354* | G/A | 0.96 (0.007) | 3.00E-08 | 0.98 (0.0119) | 0.15 | --- |
| rs7744813 | 6 | 73643289 | *KCNQ5* | A/C | 1.04 (0.0067) | 3.68E-09 | 1.02 (0.0035) | 4.38E-10 | +++ |
| rs73015318 | 6 | 163855539 | *QKI* | A/C | 1.07 (0.0089) | 1.43E-12 | 1.05 (0.0047) | 8.70E-24 | +++ |
| rs10258092 | 7 | 28547607 | *CREB5* | C/T | 1.05 (0.009) | 3.27E-08 | 1.03 (0.0047) | 1.50E-08 | +++ |
| **rs17172647** | 7 | 46214253 | *IGFBP3-TNS3* | G/A | 1.09 (0.0074) | 4.22E-29 | 1.07 (0.0038) | 5.37E-76 | +++ |
| **rs62621812** | 7 | 127015083 | *ZNF800* | A/G | 1.22 (0.0237) | 4.93E-17 | 1.12 (0.0119) | 4.86E-22 | +++ |
| rs12114462 | 8 | 22539641 | *BIN3-EGR3* | C/T | 1.04 (0.007) | 4.62E-08 | 1.02 (0.0044) | 2.45E-06 | +++ |
| rs370102919 | 8 | 103676072 | *KLF10-AZIN1* | G/GAA | 0.96 (0.0068) | 4.72E-09 | NA | NA | --NA |
| **rs1679013** | 9 | 22206987 | *CDKN2B-DMRTA1* | T/C | 1.07 (0.0071) | 1.07E-20 | 1.03 (0.0035) | 1.29E-13 | +++ |
| rs62547244 | 9 | 92050323 | *SEMA4D* | T/C | 1.04 (0.0076) | 1.08E-08 | 1.01 (0.004) | 0.082 | +++ |
| rs4742654 | 9 | 108409041 | *FKTN-TAL2* | T/G | 1.04 (0.0073) | 5.12E-09 | 1.02 (0.0038) | 7.03E-05 | +++ |
| rs4837205 | 9 | 130681892 | *ST6GALNAC4-PIP5KL1* | C/T | 0.95 (0.0094) | 2.97E-08 | 0.98 (0.0050) | 1.10E-04 | --- |
| **rs1014607** | 10 | 29024130 | *BAMBI-LINC01517* | A/G | 0.95 (0.0073) | 7.80E-11 | 0.96 (0.0039) | 2.21E-21 | --- |
| rs2274224 | 10 | 96039597 | *PLCE1* | C/G | 1.04 (0.0066) | 1.84E-08 | 1.03 (0.0034) | 5.74E-24 | +++ |
| rs779436795 | 10 | 101647971 | *DNMBP* | GTTTGTTTTGTT/G | 1.04 (0.0066) | 1.57E-09 | NA | NA | ++NA |
| **rs73386631** | 11 | 202017 | *ODF3-BET1L* | T/C | 1.13 (0.0155) | 2.81E-14 | 1.06 (0.0086) | 3.59E-11 | +++ |
| **rs150648223** | 11 | 68942162 | 5' *LOC338694* | ATTT/A | 1.17 (0.0109) | 1.12E-44 | 1.1 (0.0059) | 7.32E-60 | +++ |
| rs17739338 | 12 | 30884092 | *CAPRIN2* | T/C | 0.91 (0.0126) | 2.63E-13 | 0.94 (0.0063) | 2.64E-22 | --- |
| rs1038196 | 12 | 66343400 | *HMGA2* | C/G | 0.96 (0.0066) | 1.21E-08 | 0.99 (0.0034) | 0.001763 | --- |
| rs17608087 | 12 | 110044342 | *MVK-FAM222A* | G/A | 1.07 (0.0124) | 1.70E-08 | 1.05 (0.0067) | 1.36E-11 | +++ |
| rs7154613 | 14 | 25472083 | *STXBP6* | T/C | 1.06 (0.0097) | 1.42E-08 | 1.03 (0.0051) | 1.01E-06 | +++ |
| rs2855530 | 14 | 54421917 | *BMP4* | C/G | 1.04 (0.0066) | 1.02E-08 | 1.03 (0.0034) | 7.01E-17 | +++ |
| **rs72714121** | 15 | 28334889 | *OCA2* | T/G | 1.1 (0.0118) | 9.32E-16 | 1.05 (0.0063) | 3.47E-17 | +++ |
| rs12901945 | 15 | 61802203 | *RORA-VPS13C* | A/G | 1.05 (0.0067) | 1.91E-12 | 1.02 (0.0035) | 8.38E-08 | +++ |

**Table 1 (continued)**

| SNP | Chr | Pos | Locus | Alleles A1/A2 | Combined Meta-Analysis OR (SE) | P | Replication in 23andMe OR (SE) | P | Direction of Effect (GERA-UKB-23andMe) |
|---|---|---|---|---|---|---|---|---|---|
| rs10500355 | 16 | 7459347 | RBFOX1 | A/T | 1.05 (0.0069) | 5.47E−12 | 1.04 (0.0036) | 6.33E−26 | +++ |
| rs73530148 | 16 | 30070540 | ALDOA | T/C | 1.08 (0.0133) | 2.57E−09 | 1.04 (0.0069) | 5.02E−07 | +++ |
| **rs73568154** | **16** | **69884306** | **WWP2** | **A/T** | **1.05 (0.0066)** | **5.38E−15** | **1.02 (0.0034)** | **8.71E−09** | +++ |
| rs11150202 | 16 | 79835003 | LINC01229 | A/G | 1.04 (0.0076) | 9.63E−09 | 1.00 (0.0038) | 0.63 | +++ |
| rs8074331 | 17 | 30571412 | RHOT1 – RHBDL3 | A/T | 0.96 (0.0073) | 3.11E−08 | 0.98 (0.0038) | 7.06E−10 | −−− |
| **rs7207025** | **17** | **41520200** | **near MIR2117HG** | **A/G** | **0.96 (0.0071)** | **1.75E−08** | **0.98 (0.0036)** | **8.36E−10** | −−− |
| rs9038 | 17 | 75495397 | SEPT9 | C/T | 0.96 (0.0069) | 8.28E−09 | 0.96 (0.0035) | 8.90E−28 | −−− |
| **rs9895741** | **17** | **79603831** | **NPLOC4** | **G/A** | **0.93 (0.0084)** | **1.20E−17** | **0.95 (0.0037)** | **1.48E−42** | −−− |
| **rs75954926** | **17** | **81061048** | **3' METRNL** | **G/A** | **1.07 (0.0084)** | **2.61E−15** | **1.03 (0.0041)** | **4.23E−13** | +++ |
| rs61744414 | 19 | 17100550 | CPAMD8 | T/A | 1.15 (0.0256) | 3.47E−08 | 1.1 (0.0135) | 5.02E−13 | +++ |
| **rs549768142** | **20** | **10649677** | **JAG1** | **GAAAAAAAAAT/G** | **0.94 (0.0087)** | **3.19E−14** | **0.96 (0.0045)** | **3.34E−17** | −−− |
| **rs4814857** | **20** | **19457268** | **SLC24A3** | **G/A** | **1.16 (0.0088)** | **1.84E−61** | **1.09 (0.0047)** | **1.58E−70** | +++ |
| rs35089120 | 22 | 30531101 | HORMAD2 | C/CAT | 1.05 (0.0089) | 1.47E−08 | 1.03 (0.0047) | 9.62E−13 | +++ |

Highlighted in bold are previously reported loci (from Boutin et al.[10]).

and rs549768142 (*JAG1*), suggesting that those single variants may be the causal origin of the associations observed in their respective loci.

**Genes prioritization**. A gene-based analysis, using the VEGAS2 integrative tool[22] on 22,673 genes, found significant associations with cataract for 8 genes within 4 loci identified in the multiethnic combined (GERA + UKB) meta-analysis, including *EFNA1* and *KRTCAP2* (chr1q22), *CDKN2B* and *CDKN2B-AS1* (chr9p21.3), *MRPL21* and *LOC338694* (chr11q13.3), *HERC2* (chr15q13.2), and *BLVRA* gene (chr7p13) (Supplementary Data 9).

**Gene expression in lens tissues**. We next examined the expression of genes within identified loci potentially associated with cataract in lens tissue using the web-resource tool iSyTE (integrated Systems Tool for Eye gene discovery)[15,16]. iSyTE contains genome-wide expression data, based on microarray or RNA-seq analysis, on the mouse lens at different embryonic and postnatal stages[15,23]. In addition to expression, iSyTE also contains information of "lens-enriched expression" which has proved to be an excellent predictor of cataract-linked genes in humans and animal models[16,24–31]. The iSyTE-based lens microarray data on Affymetrix and/or Illumina platforms showed that orthologs of 47 candidates were significantly expressed in the mouse lens (>100 expression units, $P < 0.05$) in one or more embryonic/postnatal stages (Fig. 3). Over 60% of the expressed genes were found to have high lens-enriched expression (>1.5 fold-change over whole embryonic body (WB) reference dataset, $P < 0.05$), suggesting their likely relevance to lens development, homeostasis and pathology (Supplementary Fig. 9). This was further supported by iSyTE RNA-seq data that also showed lens-expression of 46 candidates ($\geq 2.0$ CPM, counts per million, $P < 0.05$), 31 out of which (~68%) exhibited high lens-enriched expression in one or more embryonic/postnatal stages (>1.5 fold-change over WB, $P < 0.05$) (Fig. 3 and Supplementary Fig. 9). Expression or lens-enriched expression heat-map for these newly identified candidate genes can be accessed through the iSyTE web-tool (https://research.bioinformatics.udel.edu/iSyTE), which allows further assessment of their expression to previously identified genes linked to cataract[15]. Together, this analysis offered strong support for lens expression for total 52 different genes, with at least one candidate gene for 43 of the 54 loci, thus accounting for nearly 80% of the identified loci. Additionally, iSyTE also informs on lens gene expression changes in specific gene perturbation mouse models that exhibit lens defects/cataract. These models were selected because of their relevance to cataract. For example, *FOXE3* mutations are linked to cataract and eye defects in human and mouse disease models[32–34], *HSF4* mutations are linked to cataract in human and mouse disease models[33,35,36], *PAX6* mutations are linked to eye defects and cataract in human and various animal models[37,38], *TDRD7* mutations are linked to cataract in human and various animal models[25,31,39–42], *Sparc* knockout mice exhibit cataract[43], *Klf4* lens-specific conditional knockout mice exhibit cataract[44], *Mafg−/−:Mafk+/−* compound mice exhibit cataract[29], *Notch2* lens-specific conditional knockout mice exhibit cataract[45], *E2f1:E2f2:E2f3* triple lens-specific conditional knockout mice exhibit cataract[46] and *Brg1* dominant negative expression in the lens results in cataract[36]. This analysis showed that 38 candidate genes had significant differences in gene expression ($P < 0.05$) in one or more of the 9 different gene perturbation mouse models of lens defects/cataract (Supplementary Fig. 10 and Supplementary Data 10). Together, iSyTE analysis offers independent experimental evidence that support the direct relevance of these candidate genes to lens biology and cataract.

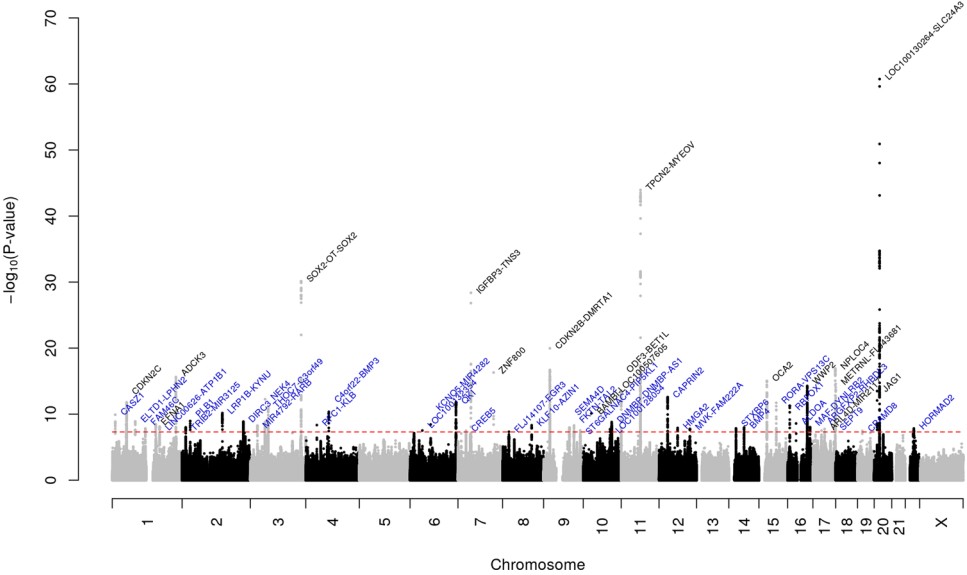

**Fig. 1 Manhattan plot of the multiethnic combined (GERA + UKB) GWAS meta-analysis of cataract.** The y-axis represents the -log₁₀(*P*-value); all *P*-values derived from logistic regression model are two-sided. The red dotted line represents the threshold of $P = 5 \times 10^{-8}$ which is the commonly accepted threshold of adjustments for multiple comparisons in GWAS. Locus names in blue are for the novel loci and the ones in dark are for the previously reported ones.

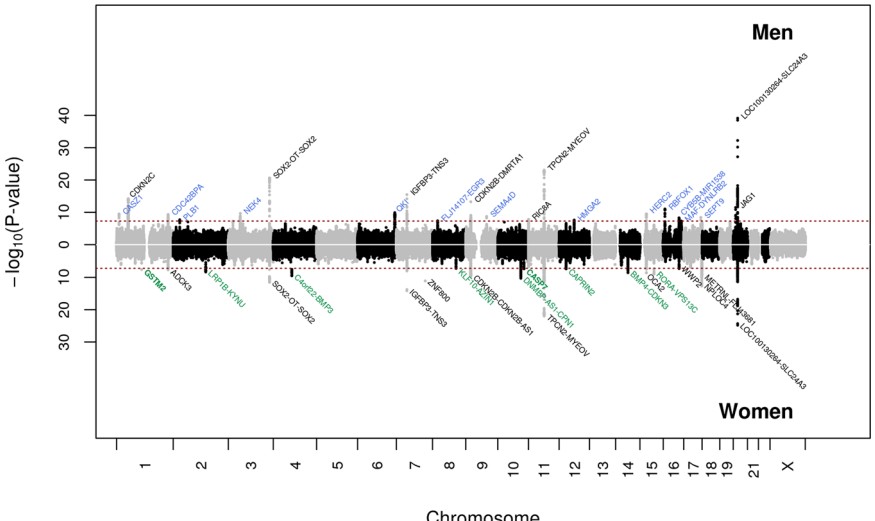

**Fig. 2 Chicago plot of the sex-stratified multiethnic GWAS meta-analyses of cataract.** Results from the meta-analysis combining men from GERA and UKB are presented on upper panel, while results from the meta-analysis combining women from GERA and UKB are presented on the lower panel. The y-axis represents the -log₁₀(*P*-value); all *P*-values derived from logistic regression model are two-sided. The red dotted line represents the threshold of $P = 5 \times 10^{-8}$ which is the commonly accepted threshold of adjustments for multiple comparisons in GWAS. Locus names in black are for those previously reported. Locus names in bold (*CASP7* and *GSTM2*) are for the additional novel loci specific to women (compared to the multiethnic meta-analysis (GERA + UKB)). Novel loci significantly associated ($P < 5 \times 10^{-8}$) with cataract in women are highlighted in green, and those significantly associated with cataract in men are highlighted in blue.

**RT-PCR validation**. It has been well established in previous studies that majority of the genes determined as "lens expressed" by iSyTE indeed prove to be expressed in the lens as examined by other methods. We sought to independently validate several GWAS-identified candidates by reverse transcriptase (RT)-polymerase chain reaction (PCR) assay for their expression in mouse lens at embryonic and postnatal stages (Supplementary Fig. 11). These data demonstrate that many candidate genes—involved in a variety of different functions—are indeed robustly expressed in the mouse lens, in turn offering further independent support for their relevance in lens tissue.

**Pathways and gene-sets enrichment**. We also conducted a pathway analysis using VEGAS software[22] to assess enrichment in 9,732 pathways or gene-sets derived from the Biosystem's database. We found that the notochord development was the only gene-set significantly enriched in our results, after correcting for multiple testing ($P < 5.14 \times 10^{-6}$) (Supplementary Data 11). This 'notochord development' gene-set consists in 18 genes, including *EPHA2*, *EFNA1*, and *NOTO*. *EPHA2* encodes the EPH receptor A2, and mutations in this gene are the cause of certain genetically-related cataract disorders, including congenital cataract and age-related cataract[47–51]. *EFNA1* encodes the ephrin A1 which has been

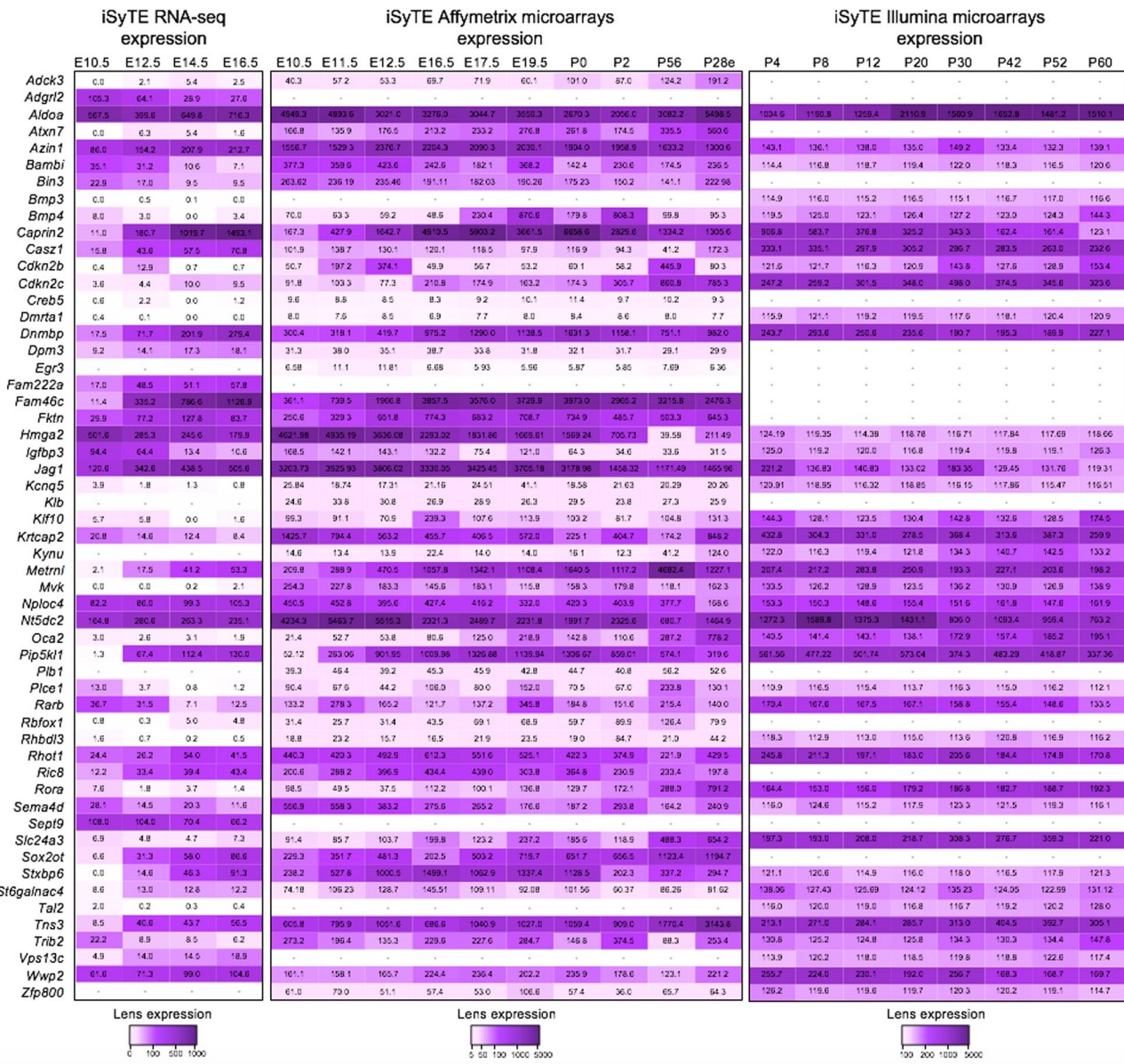

**Fig. 3 Expression of candidate genes in mouse lens.** Mouse orthologs of the human candidate genes in the 54 loci were examined for their lens expression in the iSyTE database. Analysis of whole lens tissue data on various platforms, microarrays (Affymetrix, Illumina) and RNA-seq indicates expression of 55 genes at different stages indicated by embryonic (E) and postnatal (P) days and ranged from early lens development (i.e., E10.5) through adulthood (i.e., P60). Note: P28 in Affymetrix represents expression data on isolated lens epithelium. The range of expression on each platform is indicated by a specific heat-map. The numbers within individual tiles indicate the level of expression in fluorescence intensity (for microarrays) and in counts per million (CPM) (for RNA-seq).

implicated in mediating developmental events[52] and in apoptosis and retinal epithelial development[53]. Interestingly, *EFNA1* is located within the *DPM3-KRTCAP2* cataract-associated locus identified in the current study. The *NOTO* gene encodes a homeobox that is essential for the development of the notochord in zebrafish and mouse models[54,55]. Finally, the *COL2A1* gene encodes collagen type II alpha 1 chain; mutations in this gene can cause Stickler Syndrome Type 1 which is a heterogeneous group of collagen tissue disorders, characterized by orofacial features, and ophthalmological features such as high myopia, vitreoretinal degeneration, retinal detachment, and presenile cataracts[56,57]. Future studies could clarify the relationship between genes and pathways commonly involved in notochord development and lens/cataract risk. In addition, we identified 781 pathways/gene-sets that were nominally enriched ($P < 0.05$), with the most significant of which were 'circadian clock'

($P = 2.07 \times 10^{-5}$), followed by 'lens morphogenesis in camera-type eye' ($P = 2.14 \times 10^{-5}$), and 'notochord morphogenesis' ($P = 2.81 \times 10^{-5}$). Our findings are consistent with early work, demonstrating that mice deficient in circadian clock proteins, such as *BMAL1* and *CLOCK*, display age-related cataract[58,59].

**Genome-wide genetic correlations**. To estimate the pairwise genetic correlations ($r_g$) between cataract and more than 700 diseases/traits from different publicly available resources/consortia, we compared our GWAS results with summary statistics for other traits by performing an LD score regression using the LD Hub web interface[17]. Genetic correlations were considered significant after Bonferroni adjustment for multiple testing ($P < 6.48 \times 10^{-5}$ which corresponds to 0.05/772 phenotypes tested).

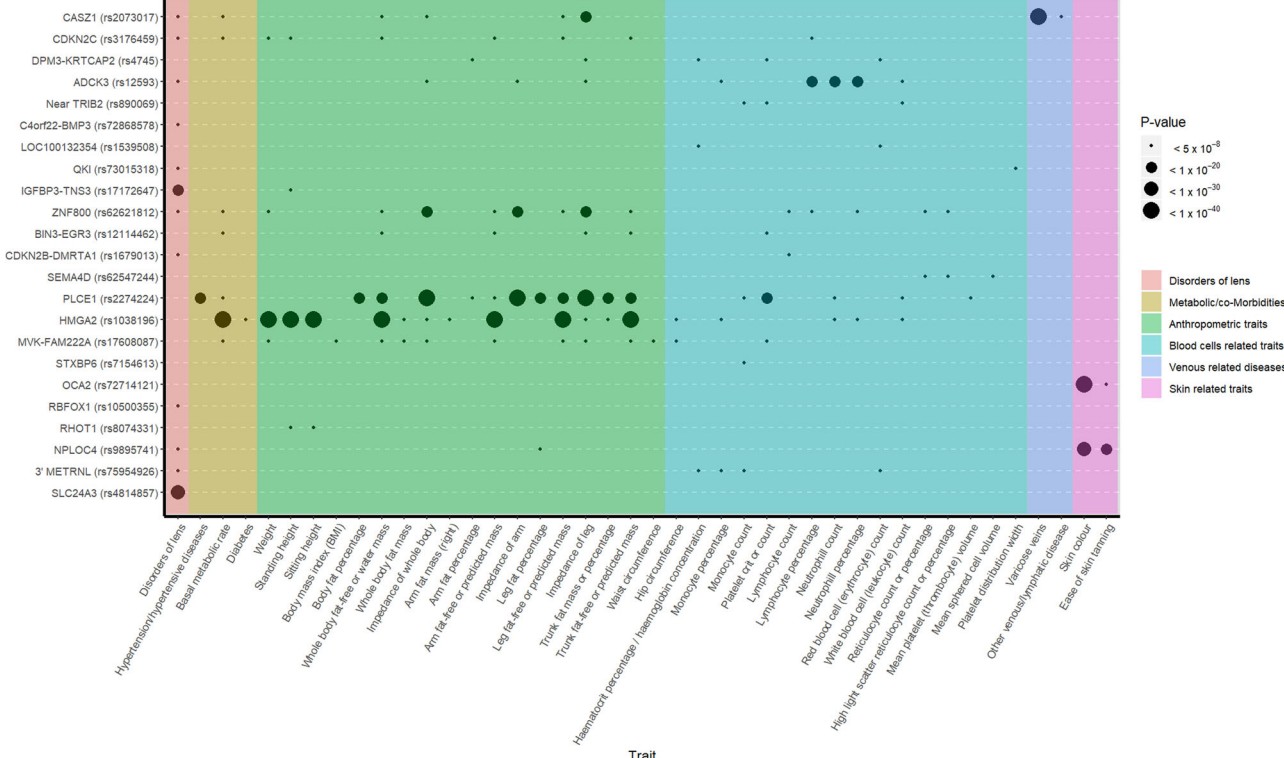

**Fig. 4 Phenome-wide association matrix of cataract top variants.** PheWAS was carried out for the 54 lead SNPs in our loci of interest identified in the combined (GERA + UKB) multiethnic analysis. SNPs were queried against 776 traits ascertained for UKB participants and reported in the Roslin Gene Atlas[60], including disorders of the lens, anthropometric traits, hematologic laboratory values, ICD-10 clinical diagnoses and self-reported conditions. Among the 54 lead SNPs, 43 were available in Gene Atlas database. We reported SNPs showing genome-wide significant association with at least one trait (in addition to cataract).

We found significant genetic correlations between cataract and 39 traits, including three of them directly related to eye traits: 'wears glasses or contact lenses' ($r_g = 0.30$, $P = 2.56 \times 10^{-7}$), 'self-reported: glaucoma' ($r_g = 0.30$, $P = 4.57 \times 10^{-6}$), and 'reason for glasses/contact lenses: myopia' ($r_g = 0.25$, $P = 1.10 \times 10^{-5}$) (Supplementary Data 12).

**Pleiotropic analyses.** A phenome-wide association study (PheWAS) analysis of 43 cataract-associated SNPs, available in the GeneATLAS was run across 776 traits measured and previously analyzed in the UKB[60]. Twenty-three of the most significantly associated cataract-associated variants were significantly associated ($P < 5.0 \times 10^{-8}$) with other traits (Fig. 4). Most were associated with disorders of the lens, with the strongest association observed for the intronic variant rs4814857 at *SLC24A3* ($P = 2.48 \times 10^{-39}$) (Supplementary Data 13). *SLC24A3* encodes the carrier family 24 member 3 and has been thought to be involved in retinal diseases[61]. Variants at *PLCE1* and *HMGA2* were significantly associated with hypertension, diabetes, and anthropometric traits, such as body fat mass and waist circumference. Although the relationship between age-related cataract and metabolic syndrome has been well established[62–65], the molecular mechanisms underlying these clinical observations remain poorly understood. Our PheWAS findings revealed that *PLCE1* and *HMGA2* could be the genetic links between age-related cataract and metabolic syndrome. Our PheWAS analysis also highlighted that variants at *OCA2* and *NPLOC4* were significantly associated with pigmentation phenotypes. The *OCA2* gene encodes the melanosomal transmembrane protein, whose variants determine iris color and have been linked to corneal and refractive astigmatism, syndromic forms of myopia, refractive error, and type 2

oculocutaneous albinism[66–70] (Supplementary Data 14). *NPLOC4* encodes the homolog, ubiquitin recognition factor and has been previously associated with macular thickness and the risk of strabismus and corneal and refractive astigmatism[67,71,72]. Despite compelling evidence, our PheWAS results raise the need of further studies to keep unraveling these complex human genome-phenome relationships and unveiling the molecular mechanisms that support them[73].

## Discussion

Our study should be interpreted within the context of its limitations. First, the cataract phenotypes were assessed differently across the 3 study cohorts. While our cataract phenotype in GERA was based on electronic health records (EHRs) data and International Classification of Disease, Ninth (ICD9) or Tenth Revision (ICD10) diagnosis codes, most of the cataract cases in UKB, and all of the cataract cases in 23andMe research cohort (our replication sample) were based on self-reported data. This may result in phenotype misclassification, however, our meta-analysis combining GERA and UKB showed consistency of the SNPs effect estimates between cohorts, and the identified associations were well validated in the 23andMe research cohort. Second, our discovery analysis mainly focused on the cataract surgery phenotype, and as cataracts generally begin to develop in people age 40 years and older, some individuals with early cataract or who will go on to develop cataract later in life might be in the control groups. However, we feel that 'cataract surgery' represents a deeper phenotype for age-related cataract and a strong validation of the diagnosis as it is conducted at a later stage of the disease. An extension of the cataract phenotypes (e.g. cataract diagnosis) investigated in GWAS is more likely to result

in the discovery of additional loci (e.g. specific to earlier stage of the disease) and could provide important biological mechanisms underlying cataract development. Subtypes of cataract were not available in the 3 study cohorts, which may result in under-estimates of the effects of individual SNPs due to phenotype misclassification. Finally, iSyTE and RT-PCR based analysis confirmed the expression of many new candidate genes in the lens. Future studies will determine whether the identified loci contribute to different cataract subtypes (i.e. nuclear, cortical, or subcapsular) and the extent to which these loci display shared effects across subtypes.

In conclusion, we report the results of a large GWAS that identified 47 novel loci (37 from the multiethnic-meta-analysis + 3 European-specific meta-analysis + 5 conditional analysis + 2 from the female-specific meta-analysis) for the development of cataract and that likely contribute to the pathophysiology of this common vision disorder. Several genes within these cataract-associated loci, including *RARB*, *KLF10*, *DNMBP*, *HMGA2*, *MVK*, *BMP4*, *CPAMD8*, and *JAG1*, represent potential candidates for the development of drug targets as previous work supports the relevance of these candidates to cataracts[74–81] (Supplementary Data 14). We also report three loci that show women-specific effects on cataract susceptibility and 4 others that showed significant differences in effects between women and men. The web tool for eye gene discovery iSyTE offers independent expression-based evidence in support of the relevance of majority of the candidate genes to lens biology and cataract. These loci provide a biological foundation for understanding the etiology of sex-differences in cataract suscept-ibility, and, may suggest potential targets for the development of non-surgical treatment of cataracts.

## Methods

**GERA.** The Genetic Epidemiology Research in Adult Health and Aging (GERA) cohort contains genome-wide genotype, clinical and demographic data of over 110,000 adult members from mainly 4 ethnic groups (non-Hispanic white, Hispanic/Latino, East Asian, and African American) of the Kaiser Permanente Northern California (KPNC) Medical Care Plan[11,82]. The Institutional Review Board of the Kaiser Foundation Research Institute has approved all study procedures. Patients with pseudophakia were diagnosed by a Kaiser Permanente ophthalmologist and were identified in the KPNC electronic health record system based on the following International Classification of Disease, Ninth (ICD9) or Tenth Revision (ICD10) diagnosis codes: V43.1 (ICD-9 code) and Z96.1 (ICD-10 code). Cataract cases were also identified if they had a history of having a cataract surgery at KPNC. Our control group included all the non-cases. In total, 33,145 patients who have undergone cataract surgery and 64,777 controls from GERA were included in this study.

Protocols for participant genotyping data collection and previous quality control have been described in detail[82]. Briefly GERA participants' DNA samples were extracted from Oragene kits (DNA Genotek Inc., Ottawa, ON, Canada) at KPNC and genotyped at the Genomics Core Facility of UCSF. DNA samples were genotyped at over 665,000 genetic markers on four ethnic-specific Affymetrix Axiom arrays (Affymetrix, Santa Clara, CA, USA) optimized for European, Latino, East Asian, and African American individuals[83,84]. Genotype quality control (QC) procedures and imputation were conducted on an array-wise basis[82], after an updated genotyping algorithm with an advanced normalization step specifically for SNPs in batches not recommended or flagged by the outlier plate detector than has previously been done. Subsequently, variants were excluded if: >3 clusters were identified; the number of batches was <38/42 (EUR array), <3/5 (AFR), < 3/6 (EAS), or <7/9 (LAT); and the ratio of expected allele frequency variance across packages was <100 (EUR), < 50 (AFR), < 100 (EAS), < 200 (LAT). On the EUR array, variants were additionally excluded if heterozygosity > .52 or < .02, and if an association test between Reagent kit v1.0 and v2.0 had $P < 10^{-4}$. Imputation was done by array, and we additionally removed variants with call rates <90%. Genotypes were then pre-phased with Eagle[85] v2.3.2, and then imputed with Minimac3[86] v2.0.1, using two reference panels. Variants were preferred if present in the EGA release of the Haplotype Reference Consortium (HRC; $n = 27,165$) reference panel[87], and from the 1000 Genomes Project Phase III release if not ($n = 2504$; e.g., indels)[88].

We first analyzed each ethnic group (non-Hispanic white, Hispanic/Latino, East Asian, and African American) separately. We ran a logistic regression of cataract and each SNP using PLINK[89] v1.9 (www.cog-genomics.org/plink/1.9/) adjusting for age, sex, and ancestry principal components (PCs), which were previously[11] assessed within each ethnic group using Eigenstrat[90] v4.2. We included as covariates the top ten ancestry PCs for the non-Hispanic whites, whereas we included the top six ancestry PCs for the three other ethnic groups. To adjust for genetic ancestry, we also included the percentage of Ashkenazi (ASHK) ancestry as a covariate for the non-Hispanic white sample analyses[11].

**UK Biobank.** The UK Biobank(UKB) is a large prospective study following the health of approximately 500,000 participants from 5 ethnic groups (European, East Asian, South Asian, African British, and mixed ancestries) resident in the UK aged between 40 and 69 years-old at the baseline recruitment visit[13,91]. Demographic information and medical history were ascertained through touch-screen questionnaires. Participants also underwent a wide range of physical and cognitive assessments, including blood sampling. Cataract cases ($N = 34,699$) were defined as participants with a self-reported cataract operation (f20004 code 1435) or/and a hospital record including a diagnosis code (ICD-10: H25 or H26). Controls ($N = 452,622$) were participants who were not cases. Phenotyping, genotyping and imputation were carried out by members of the UK Biobank team. Imputation to the Haplotype Reference Consortium reference panel plus the 1000 Genomes Project and UK10K reference panels has been described (www.ukbiobank.ac.uk). Following QC, over 10 million variants in 487,622 individuals were tested for cataract adjusting for age, sex, and genetic ancestry principal components.

GWAS analysis was performed by ethnic group. Ethnic groups were formed by those who reported any white group and with global ancestry $PC_1 \le 50$ and $PC_2 \le 50$ (where global $PC_1$ and $PC_2$ were calculated from the entire cohort), and by those reporting East Asian, South Asian, African British, and mixed/other ancestries. Ancestry PCs were then calculated within each ethnic group as done in GERA[11], using 50,000 random individuals and the rest projected just for Europeans, and GWAS analysis adjusted for 10 PCs in all ethnic groups. The analyses presented in this paper were carried out under UK Biobank Resource project #14105.

**GWAS meta-analyses.** First, a meta-analysis of cataract was conducted in GERA to combine the results of the 4 ethnic groups using the R[92] (https://www.R-project.org) package "meta". Similarly, a meta-analysis was conducted in UKB to combine the results of the 5 ethnic groups. Three ethnic-specific meta-analyses were also performed: (1) combining European-specific samples (i.e., GERA non-Hispanic whites and UKB Europeans); (2) combining East Asian-specific samples (i.e. GERA and UKB East Asians); and (3) combining African-specific samples (i.e. GERA African Americans and UKB Africans). A meta-analysis was then conducted to combine the results from GERA and UKB. Two sex-specific meta-analyses were also performed: (1) combining women from GERA and UKB; and (2) combining men from GERA and UKB. Fixed effects summary estimates were calculated for an additive model. We also estimated heterogeneity index, $I^2$ (0–100%) and $P$-value for Cochrane's Q statistic among different groups, and studies. For each locus, we defined the top SNP as the most significant variant within a 2 Mb window. Novel loci were defined as those that were located over 1 Mb apart from any previously reported locus[10].

**Conditional & joint (COJO) analysis.** A multi-SNP-based conditional & joint association analysis (COJO)[21] was performed on the combined European-specific (GERA non-Hispanic whites + UKB Europeans) meta-analysis results to potentially identify independent signals within the 44 identified genomic regions. To calculate linkage disequilibrium (LD) patterns, we used 10,000 randomly selected samples from GERA non-Hispanic white ethnic group as a reference panel. A $P$-value less than $5.0 \times 10^{-8}$ was considered as the significance threshold for this COJO analysis.

**Replication in 23andMe.** Replication analysis of 54 loci identified in the combined (GERA + UKB) meta-analysis as well as the sex-stratified association signals identified in the women- or men-specific meta-analysis (GERA + UKB) was conducted using self-reported data from a GWAS including 347,209 self-reported cataract cases and 2,887,246 controls (close relatives removed) of 5 ethnic groups (i.e., European, Latino, East Asian, South Asian, and African American) determined through an analysis of local ancestry[93], from 23andMe, Inc., research cohort. Participants provided informed consent and participated in the research online, under a protocol approved by the external AAHRPP-accredited IRB, Ethical & Independent Review Services (E&I Review). The self-reported phenotype was derived from survey questions. Cases were defined as those individuals that reported having cataract whereas controls were defined as individuals that reported not having cataract. Individuals that preferred not to/did not answer the cataract questions were excluded from the analysis. In 23andMe replication analysis, a maximal set of unrelated individuals was chosen for each analysis using a segmental identity-by-descent (IBD) estimation algorithm. Individuals were defined as related if they shared more than 700 cM IBD, including regions where the two individuals share either one or both genomic segments IBD. When selecting individuals for case/control phenotype analyses, the selection process is designed to maximize case sample size by preferentially retaining cases over controls. Specifically, if both an individual case and an individual control are found to be related, then the case is retained in the analysis. Variant QC is applied independently to genotyped and imputed GWAS results. The SNPs failing QC are flagged based on multiple criteria, such as Hardy-Weinberg P-value, call rate, imputation R-square and test statistics of batch effects. Analyses were carried out through logistic regression assuming an additive model for allele effects and adjusting for age, sex, indicator variables to represent the genotyping platforms and the first five genotype principal components.

**Variants prioritization**. To prioritize variants within the 54 identified genomic regions for follow-up functional evaluation, a Bayesian approach (CAVIARBF)[14] was used, which is available publicly at https://bitbucket.org/Wenan/caviarbf. Each variant's capacity to explain the identified signal within a 2 Mb window (±1.0 Mb with respect to the original top variant) was computed for each identified genomic region. Then, the smallest set of variants that included the causal variant with 95% probability (95% credible set) was derived. Out of the 1359 total variants, 43 variants had > 20% probability of being causal.

**VEGAS2 prioritization**. To prioritize genes and biological pathways, we conducted a gene-based and pathways analyses using the Versatile Gene-based Association Study - 2 version 2 (VEGAS2v02) web platform[22]. We first performed a gene-based association analysis on the combined (GERA + UKB) meta-analysis results using the default '-top 100' test that uses all (100%) variants assigned to a gene to compute gene-based $P$-value. Gene-based analyses were conducted on each of the individual ethnic groups (European-specific samples (GERA and UKB individuals), GERA Hispanic/Latinos, East Asian-specific samples (GERA and UKB individuals), UKB South Asians, and GERA African Americans) using the appropriate reference panel: 1000 Genomes phase 3 European population, 1000 Genomes phase 3 American population, 1000 Genomes phase 3 East and South Asian populations, and 1000 Genomes phase 3 African population, respectively. We then meta-analyzed the 5 ethnic groups gene-based results using Fisher's method for combining the $P$-values. As 22,673 genes were tested, the P-value adjusted for Bonferroni correction was set as $P < 2.21 \times 10^{-6}$ (0.05/22,673).

Second, we performed pathways analyses based on VEGAS2 gene-based $P$-values. We tested enrichment of the genes defined by VEGAS2 in 9,732 pathways or gene-sets (with 17,701 unique genes) derived from the Biosystem's database (https://vegas2.qimrberghofer.edu.au/biosystems20160324.vegas2pathSYM). We adopted the resampling approach to perform pathway analyses using VEGAS2 derived gene-based $P$-value considering the default '−10 kbloc' parameter as previously used[94]. We then meta-analyzed the 5 ethnic groups gene-based results using Fisher's method for combining the $P$-values. As 9,732 pathways or gene-sets from the Biosystem's database were tested, the $P$-value adjusted for Bonferroni correction was set as $P < 5.14 \times 10^{-6}$ (0.05/9,732).

**iSyTE analyses for lens gene expression**. The iSyTE database was used to analyze mouse orthologs of the human candidate genes in the 54 loci linked to cataract. iSyTE contains genome-wide transcript expression information on mouse lens obtained from microarrays and RNA-sequencing (RNA-seq) studies[15,23]. The Affymetrix 430 2.0 platform (GeneChip Mouse Genome 430 2.0 Array and/or 430 A 2.0 Array) data used in this analysis was obtained on mouse whole lens tissue at embryonic day (E) stages E10.5, E11.5, E12.5, E16.5, E17.5, E19.5, as well as postnatal (P) day stages P0, P2, and P56, in addition to isolated lens epithelium at P28. The Illumina platform (BeadChip MouseWG-6 v2.0 Expression arrays) data used in this analysis was obtained on mouse whole lens tissue at P4, P8, P12, P20, P30, P42, P52, and P60. Because previously we have shown that lens-enriched expression of a candidate gene can be used as indicative of its potential function in the lens[15,16], we also examined the lens-enrichment of the candidate genes. This was evaluated as elevated expression in the lens compared to that in whole mouse whole embryonic body (WB)-based on a previously reported WB-in silico subtraction approach[15,16,95]. In brief, microarray files were imported in the R statistical environment (http://www.r-project.org), and processed using relevant packages implemented in Bioconductor v3.12 (https://www.bioconductor.org). Probe sets were further processed to derive present/absent calls and further by *limma* to collapse into genes, based on significant *p*-values and highest median expression. Comparative analysis was performed in *limma* using lmFit and makeContrasts functions to identify differential expression of genes in the lens datasets compared to WB datasets. Expression of candidate genes was also examined in RNA-seq data from wild-type mouse whole lenses at stages E10.5, E12.5, E14.5 and E16.5 obtained in a previous study[23].

**Expression analyses in specific gene-perturbation mouse models of lens defects/cataract**. The iSyTE database was also used to examine expression of mouse orthologs of the candidate genes in the context of ten different gene perturbation conditions in transgenic, mutant, or targeted knockout mouse models that exhibit lens defects and/or cataract. The following mouse lens gene expression microarray data were analyzed: *Brg1* dominant negative dnBrg1 transgenic mice at E15.5 (GSE22322) (four biological replicates for control and transgenic animals), *E2f1:E2f2:E2f3* conditional lens-specific triple targeted knockout mice at E17.5 and P0 (GSE16533) (five biological replicates for control and triple knockouts at E17.5 and P0), *Foxe3* Cryaa-promoter-driven lens over-expression transgenic mice at P2 (GSE9711) (three biological replicates for control and transgenic animals), *Hsf4* germline targeted knockout mice at P0 (GSE22362) (three biological replicates for control and knockout animals), *Klf4* conditional lens-specific targeted knockout mice at E16.5 and P56 (GSE47694) (three biological replicates for control and knockout animals at E16.5 and two biological replicates for control and knockout animals at P56), *Mafg−/−:Mafk+/−* compound germline targeted knockout mice at P60 (GSE65500) (two biological replicates for control and compound animals), *Notch2* conditional lens-specific targeted knockout mice at E19.5 (GSE31643) (three biological replicates for control and lens-specific knockout animals), *Pax6*

germline heterozygous targeted knockout mice at P0 (GSE13244) (three biological replicates for control and heterozygous animals), *Tdrd7* germline null (*Tdrd7^Grm5^*) mice at P30 (GSE25776) (three biological replicates for control and mutant animals), *Sparc* germline targeted knockout mice (isolated lens epithelium) at P28 (GSE13402) (three biological replicates for control and four biological replicates for knockout animals). Candidate genes were analyzed for significant differential expression in the lens ($P$-value ≤ 0.05) in one or more of the above gene-perturbation conditions and plotted in the graphs.

**Mouse lens RNA isolation and RT-PCR analysis**. Mice were maintained at the University of Delaware Animal Facility and all animal-related experimental protocols were designed according to guidelines from the Association for Research in Vision and Ophthalmology (ARVO) statement for the use of animals in ophthalmic and vision research. The University of Delaware Institutional Animal Care and Use Committee (IACUC) reviewed and approved the animal protocol. Mice of C57BL/6 J strain (Taconic Biosciences) were bred and day of observation of vaginal plug was designated as embryonic day (E) 0.5 and postnatal days were designated with "P". Lens were dissected at stages E16.5 and P3 and used for isolation of total RNA using RNAeasy kit (Qiagen, Hilden, Germany, Qiagen #74104). Total RNA was used for preparation of cDNA using iScript cDNA synthesis kit (Bio-Rad #1708890EDU). Primers were designed for candidate genes (Supplementary Data 15) for RT-PCR analysis, which was performed on E16.5 and P0 cDNA using the following PCR conditions: 94 °C for 2 min, 94 °C for 30 s, 57 °C for 30 s, 72 °C for 30 s, cycled 34 times (except for housekeeping control Actb, 28 cycles), final extension at 72 °C for 7 min. The amplified PCR products were separated on a 1.5% agarose gel and imaged with UVP GelDoc-It 310 Imager (Upland, California) (Supplementary Fig. 11). Our previous findings have shown that fluorescence expression intensity units of around 100 (with significant expression p-value) in the Affymetrix and Illumina microarray platforms has served as good indicators that a candidate gene will be validated by independent assays such as RT-PCR[96,97].

**Genetic correlations**. To estimate the genetic correlation of cataract with more than 700 diseases/traits, including vision disorders, from different publicly available resources/consortia, we used the LD Hub web interface[17], which performs automated LD score regression. In the LD Score regressions, we included only HapMap3 SNPs with MAF > 0.01. Genetic correlations were considered significant after Bonferroni adjustment for multiple testing ($P < 6.48 \times 10^{-5}$ which corresponds to 0.05/772 phenotypes tested).

**PheWAS analyses**. PheWAS was carried out for the 54 lead SNPs in our loci of interest identified in the combined (GERA + UKB) multiethnic analysis. SNPs were queried against 776 traits ascertained for UKB participants and reported in the Roslin Gene Atlas[60], including disorders of the lens, anthropometric traits, hematologic laboratory values, ICD-10 clinical diagnoses and self-reported conditions. Among the 54 lead SNPs, 43 were available in Gene Atlas database. We reported SNPs showing genome-wide significant association with at least one trait (in addition to cataract).

**Reporting summary**. Further information on research design is available in the Nature Research Reporting Summary linked to this article.

## Data availability
The GERA genotype data are available upon application to the KP Research Bank (https://researchbank.kaiserpermanente.org/). The combined (GERA + UKB) meta-analysis GWAS summary statistics are available from the NHGRI-EBI GWAS Catalog (https://www.ebi.ac.uk/gwas/downloads/summary-statistics), study accession number GCST90014268. The variant-level data for the 23andMe replication dataset are fully disclosed in the manuscript (Table 1, Supplementary Data 6 and 7). Individual-level data are not publicly available due participant confidentiality, and in accordance with the IRB-approved protocol under which the study was conducted. Expression or lens-enriched expression heat-map for candidate genes can be accessed through the iSyTE web-tool (https://research.bioinformatics.udel.edu/iSyTE). Pathways or gene-sets were derived from the Biosystem's database which can be accessed through the following link (https://vegas2.qimrberghofer.edu.au/biosystems20160324.vegas2pathSYM).

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

## Acknowledgements

We are grateful to the Kaiser Permanente Northern California members who have generously agreed to participate in the Kaiser Permanente Research Program on Genes, Environment, and Health. Support for participant enrollment, survey completion, and biospecimen collection for the RPGEH was provided by the Robert Wood Johnson Foundation, the Wayne and Gladys Valley Foundation, the Ellison Medical Foundation, and Kaiser Permanente Community Benefit Programs. Genotyping of the GERA cohort was funded by a grant from the National Institute on Aging, National Institute of Mental Health, and National Institute of Health Common Fund (RC2 AG036607). H.C. and E.J. were supported by the National Eye Institute (NEI) grant R01 EY027004, the National Institute of Diabetes and Digestive and Kidney Diseases (NIDDK) R01 DK116738 and by the National Cancer Institute (NCI) R01 CA241623. This work was also made possible in part by NIH-NEI EY002162—Core Grant for Vision Research, by the Research to Prevent Blindness Unrestricted Grant (UCSF, Ophthalmology). K.S.N. receives support from NEI grant EY022891, BrightFocus Foundation (G2019360), Marin Community Foundation-Kathlyn McPherson Masneri and Arno P. Masneri Fund, and That Man May See Inc. T.J.H. was supported by National Institutes of Aging (NIA) grant R21 AG046616. S.A.L. was supported by National Institutes of Health / National Eye Institute grants R01 EY021505 and EY029770 and D.A. was supported by a Knights Templar Pediatric Ophthalmology Career Starter Grant Award. We would like to thank the research participants and employees of 23andMe for making this work possible (see Supplementary Information for the full list of members of the 23andMe Research Team).

## Author contributions

H.C., S.A.L. and E.J. contributed to study conception and design. T.J.H. and E.J. were involved in the genotyping and quality control of the GERA samples. T.J.H. performed the imputation analyses in the GERA cohort. R.B.M. extracted phenotype data for the GERA subjects based on EHRs. J.Y. performed statistical analyses and in silico analyses. D.A. and S.A.L. carried out the gene expression analyses in lens tissue using iSyTE and RT-PCR validation analyses. W.W. and G.C.P. performed the analyses pertaining to GWAS replication, and reviewed and revised the manuscript. H.C., S.A.L. and E.J. interpreted the results of analyses and wrote the manuscript with help from R.B.M., K.S.N. and P.G.H.

## Competing interests

G.C.P. and W.W. are employed by and hold stock or stock options in 23andMe, Inc. The remaining authors declare no competing interests.
