## [Peer Review File · Nature Communications]

Reviewers' Comments:

Reviewer #1:

Remarks to the Author:

Dr Choquet and co-authors present a very large multiethnic GWAS meta-analysis of 585,243 individuals ((67,844 cases and 517,399 cataract-free controls) and discovered multiple new genetic loci that are associated with susceptibility to cataract. They applied a very stringent threshold of $P < 5 \times 10^{-8}$ to bring forward loci for replication in the independently ascertained 23andME research cohort (comprising 347,209 self-reported cataract cases and 2,887,246 controls). The primary specified endpoint was rigorously analysed; although the authors state up front that the cataract phenotype ascertainment criteria varied across GERA (electronic health records), UKBB (self report), and 23andME (self report), i believe that this is a good faith attempt to best capture information that can be obtained. The consistency of the association signals across the discovery and independent replication collections, together with their large sample sizes, argues against ascertainment bias being the reason of the reported association signals. It was a great pleasure reading this manuscript. Because cataract is such a common affliction in the elderly, this manuscript would be an excellent resource of genetic information to interested readers of Nature Communications and Nature Genetics.

I had the opportunity to review the secondary endpoints and analyses, and I have some questions that may result in a markedly improved paper if they can be addressed.

1. The Sex-stratified association signals were only reported in GERA and UKBB, and were not subject to replication in 23andMe. Yet, rather strong statements about this secondary endpoint were made in the manuscript main text. I will be grateful for the authors' comments on this.
2. The primary endpoint identified 54 loci. This is a very rich harvest of genetic and potential biological insights. The secondary pathway analysis revealed "notochord development" as the only gene set that is significant after multiple testing correction. It would be very helpful for the authors to reconcile this, as it is not intuitive to the general readership (notochord = developmental = young people. Cataract = old people, mostly).
3. The iSyTE resource comprises microarray data and RNA sequencing data. This is very helpful. Can the authors consider RT-PCR validation of perhaps the most insightful findings in independent tissues? Or at least comment? This is because RNA seq data expressed in counts per million may not be very intuitive to other researchers who may like to 'visualize' the expression output.
4. The authors examined changes in the expression of candidate genes in associated loci in 9 gene perturbation mouse models of lens defects. Are these 9 mouse models relevant given the biological context (human loci, mouse expression); why were these mouse models chosen?

I hope you will find the comments helpful

All the best

Khor Chiea Chuen

Reviewer #2:

Remarks to the Author:

This manuscript describes a genome-wide association study for age-related cataract using multiple biobank cohorts. The authors identify 37 novel loci and 17 known loci. Of the 54 loci, 45 (29 novel and 16 known) were replicated in an independent cohort. The authors then stratify by ethnicity and identify an additional 3 loci, and performed conjoint analyses of loci identified in Europeans, revealing 5 more independent signals. Sex-specific analyses identified two more novel loci detected in females

only and revealed sex-based differences in effect size at 5 loci from the main meta-analysis. Gene and pathway-based analyses were used to further explore the data, along with interrogation of candidate genes at each locus in lens specific (mouse) expression data. Finally, a PheWAS approach identifies correlations with a range of other phenotypes.

The paper is clearly written, the methods are comprehensive, and the results are clearly displayed.

Specific questions:

1. Table 1: the loci in grey highlight are previously reported. Can you please indicate where they were previously reported? If they are from the paper "Boutin et al. Insights into the genetic basis of retinal detachment" it should be clearly noted that the UK Biobank cohort was used in both studies and they are not independent replications. Conversely, if there is an independent replication of a previously reported locus, that should be noted too.
2. Supplementary figure 2 indicates a lambda value of 1.139. This suggests some degree of inflation. How was this combatted and what considerations should be made for this in the final interpretation?
3. The phenotype in GERA and to some extent in UK Biobank is defined as cataract surgery (not cataract diagnosis). This means that there will be people with cataract in the control cohort. Please consider this when discussing the limitations.
4. In all the cohorts, the cases are significantly older than the controls. The controls are not old enough on average to exclude cataract. Was age adjusted for in the GWAS analyses of UKBiobank? (It is mentioned for GERA and 23andMe). Please also consider this age difference in the limitations, particularly in the context of the 23andMe replication cohort where the mean age of controls is only 45 years. Given the size of this cohort, it would be expected to be substantially more powerful than the discovery cohort, however the potential for misclassification is high (given self-report and young age).
5. How were candidate genes at each locus selected for the subsequent iSyTE analyses?
6. Please expand on the significance of the PheWAS results. Is there likely to be a biological link to cataract, or are there likely to be bias or confounding factors in phenotyping that lead to these correlations?

REVIEWER COMMENTS

Reviewer #1 (Remarks to the Author):

Dr Choquet and co-authors present a very large multiethnic GWAS meta-analysis of 585,243 individuals ((67,844 cases and 517,399 cataract-free controls) and discovered multiple new genetic loci that are associated with susceptibility to cataract. They applied a very stringent threshold of $P < 5 \times 10^{-8}$ to bring forward loci for replication in the independently ascertained 23andMe research cohort (comprising 347,209 self-reported cataract cases and 2,887,246 controls). The primary specified endpoint was rigorously analysed; although the authors state up front that the cataract phenotype ascertainment criteria varied across GERA (electronic health records), UKBB (self report), and 23andMe (self report), i believe that this is a good faith attempt to best capture information that can be obtained. The consistency of the association signals across the discovery and independent replication collections, together with their large sample sizes, argues against ascertainment bias being the reason of the reported association signals. It was a great pleasure reading this manuscript. Because cataract is such a common affliction in the elderly, this manuscript would be an excellent resource of genetic information to interested readers of Nature Communications and Nature Genetics.

I had the opportunity to review the secondary endpoints and analyses, and I have some questions that may result in a markedly improved paper if they can be addressed.

We thank the reviewer for the positive feedback and the helpful comments.

1. The Sex-stratified association signals were only reported in GERA and UKBB, and were not subject to replication in 23andMe. Yet, rather strong statements about this secondary endpoint were made in the manuscript main text. I will be grateful for the authors' comments on this.

This is a good point. We now provide replication in the 23andMe research cohort of the sex-stratified association signals identified in the women- or men-specific meta-analysis (GERA+UKB). We have reported those 23andMe replication results in **Supplementary Tables 6-7**.

Importantly, we validated the women-specific locus *CASP7* rs12777332; and we also observed significant sex differences in the effect sizes and significance of association for the following loci: *SEMA4D*, *RBFOX1*, and *JAG1* as reported in the table below:

Sex-specificity	Locus	SNP	Replication in 23andMe Research Cohort			
			Meta-analysis Female		Meta-analysis Male	
			OR (95% CI)	P	OR (95% CI)	P
Women-specific loci	CASP7	rs12777332	1.04 (1.03-1.05)	4.53×10^{-15}	1.01 (1.00-1.02)	0.14
	GSTM2	rs3819350	1.01 (1.003-1.02)	7.95×10^{-3}	1.01 (1.002-1.02)	0.025
	DNMBP-CPN1	rs1986500	0.98 (0.97-0.99)	6.01×10^{-5}	0.99 (0.98-0.997)	0.012
Men-specific loci	QKI	6:163840336	NA	NA	NA	NA
	SEMA4D	rs62547232	1.0 (0.98-1.02)	0.97	1.03 (1.01-1.06)	0.017
	RBFOX1	rs7184522	1.03 (1.02-1.04)	6.28×10^{-11}	1.05 (1.04-1.06)	2.16×10^{-17}
	JAG1	rs3790163	0.98 (0.97-0.99)	4.35×10^{-3}	0.95 (0.94-0.97)	4.93×10^{-8}

We have added some text in the Results section to reflect these 23andMe replication results, as below:

“We identified two additional novel loci, *CASP7* and *GSTM2*, in the women-specific meta-analysis (GERA+UKB) (**Supplementary Table 5**). *CASP7* rs12777332 and *GSTM2* rs3819350 were significantly associated with cataract in women (*CASP7* rs12777332: OR=1.06, $P=3.41 \times 10^{-8}$; *GSTM2* rs3819350: OR=1.06, $P=2.10 \times 10^{-8}$) but not in men (*CASP7* rs12777332: OR=1.01, $P=0.25$; *GSTM2* rs3819350: OR=1.01, $P=0.25$). While we confirmed the women-specific association at the *CASP7* locus in the 23andMe replication cohort, the sex-specific association at the *GSTM2* was not

validated (**Supplementary Table 6**). Further, among the loci identified in the multiethnic meta-analysis (GERA+UKB), we observed significant differences in the effect sizes and significance of association at five loci: one locus, *DNMBP-CPN1*, was strongly associated with cataract in women but not in men (*DNMBP-CPN1* rs1986500, OR=0.94, $P=5.04 \times 10^{-11}$ in women, and OR=1.01, $P=0.40$ in men; $Z=-5.03$, $P=2.44 \times 10^{-7}$) (**Supplementary Table 5**); and four loci, *QKI*, *SEMA4D*, *RBFOX1*, and *JAG1*, were strongly associated in men than women (*QKI* 6:163840336, OR=0.94, $P=1.23 \times 10^{-10}$ in men, and OR=0.99, $P=0.21$ in women; $Z=-3.95$, $P=3.89 \times 10^{-5}$; *SEMA4D* rs62547232, OR=1.15, $P=1.83 \times 10^{-9}$ in men, and OR=0.98, $P=0.33$ in women; $Z=5.03$, $P=2.43 \times 10^{-7}$; *RBFOX1* rs7184522, OR=1.07, $P=9.10 \times 10^{-12}$ in men, and OR=1.03, $P=0.0020$ in women; $Z=2.98$, $P=1.43 \times 10^{-3}$; *JAG1* rs3790163, OR=0.92, $P=3.14 \times 10^{-12}$ in men, and OR=0.96, $P=9.63 \times 10^{-4}$ in women; $Z=-2.95$, $P=1.59 \times 10^{-3}$) (**Supplementary Table 6**). Similarly, we observed significant sex differences in the effect sizes and significance of association in the 23andMe replication cohort for the following loci: *SEMA4D*, *RBFOX1*, and *JAG1*."

2. The primary endpoint identified 54 loci. This is a very rich harvest of genetic and potential biological insights. The secondary pathway analysis revealed "notochord development" as the only gene set that is significant after multiple testing correction. It would be very helpful for the authors to reconcile this, as it is not intuitive to the general readership (notochord = developmental = young people. Cataract = old people, mostly).

This is a very good point raised by the reviewer. By conducting a pathway analysis using VEGAS software to assess enrichment in 9,732 pathways or gene-sets derived from the Biosystem's database, we found that the notochord development was the only gene-set significantly enriched in our results, after correcting for multiple testing ($P < 5.14 \times 10^{-6}$). The exact gene-set term reported in the VEGAS pathway/gene-set analysis results is "500198_GO:0030903_notochord_development" (see **Supplementary Table 11**) and consists in 18 genes, including *EPHA2*, *EFNA1*, *NOTO* and *COL2A1*. We have collected information on those 18 genes in the NCBI Gene database as reported in the table below:

Gene	Full Name	Relevant Role/Function	Location	Gene located within a cataract-associated locus ? If yes, locus name
EPHA2	EPH receptor A2	Mutations in this gene are the cause of certain genetically-related cataract disorders	1p36.13	no
ID3	inhibitor of DNA binding 3, HLH protein	The protein encoded by this gene is a helix-loop-helix (HLH) protein that can form heterodimers with other HLH proteins.	1p36.12	no
STIL	STIL centriolar assembly protein		1p33	no
EFNA1	ephrin A1	The ephrins and EPH-related receptors comprise the largest subfamily of receptor protein-tyrosine kinases and have been implicated in mediating developmental events, especially in the nervous system and in erythropoiesis. Apoptosis and retinal epithelial development.	1q22	DPM3-KRTCAP2
NOTO	notochord homeobox		2p13.2	no
GLI2	GLI family zinc finger 2	It is also thought to play a role during embryogenesis. The encoded protein is associated with several phenotypes- Greig cephalopolysyndactyly syndrome, Pallister-Hall syndrome, preaxial polydactyly type IV, postaxial polydactyly types A1 and B.	2q14.2	no
NCKAP1	NCK associated protein 1		2q32.1	no
WNT5A	Wnt family member 5A	This protein plays an essential role in regulating developmental pathways during embryogenesis. Mutations in this gene are the cause of autosomal dominant Robinow syndrome.	3p14.3	no
TBXT	T-box transcription factor T	The protein encoded by this gene is an embryonic nuclear transcription factor that binds to a specific DNA element, the palindromic T-site. It binds through a region in its N-terminus, called the T-box, and effects transcription of genes required for mesoderm formation and differentiation. The protein is localized to notochord-derived cells. Variation in this gene was associated with susceptibility to neural tube defects and chordoma. A mutation in this gene was found in a family with sacral agenesis with vertebral anomalies. Presence of bilateral inferior altitudinal visual field defects.	6q27	no
COBL	Cordon-bleu WH2 repeat protein		7p12.1	no

CRB2	Crumbs cell polarity complex component 2	In mammals, members of this family are thought to play a role in many cellular processes in early embryonic development.	9q33.3	no
WNT11	Wnt family member 11	Wnt family member proteins have been implicated in oncogenesis and in several developmental processes, including regulation of cell fate and patterning during embryogenesis.	11q13.5	no
YAP1	Yes1 associated transcriptional regulator	This gene encodes a downstream nuclear effector of the Hippo signaling pathway which is involved in development, growth, repair, and homeostasis.	11q22.1	no
COL2A1	collagen type II alpha 1 chain	This gene encodes the alpha-1 chain of type II collagen, a fibrillar collagen found in cartilage and the vitreous humor of the eye. Mutations in this gene are associated with achondrogenesis, chondrodysplasia, early onset familial osteoarthritis, SED congenita, Langer-Saldino achondrogenesis, Kniest dysplasia, Stickler syndrome type I, and spondyloepimetaphyseal dysplasia Strudwick type. In addition, defects in processing chondrocalcin, a calcium binding protein that is the C-propeptide of this collagen molecule, are also associated with chondrodysplasia.	12q13.11	no
GLI1	GLI family zinc finger 1	PMID: 25143588	12q13.3	no
NOG	Noggin	The protein appears to have pleiotropic effect, both early in development as well as in later stages.	17q22	no
SOX9	SRY-box transcription factor 9		17q24.3	no
TEAD2	TEA domain transcription factor 2		19q13.33	no

We have added some text in the results section of the manuscript to help the readers to reconcile this significant gene-set enrichment with cataract, as follows:

“We also conducted a pathway analysis using VEGAS software to assess enrichment in 9,732 pathways or gene-sets derived from the Biosystem’s database. We found that the notochord development was the only gene-set significantly enriched in our results, after correcting for multiple testing ($P < 5.14 \times 10^{-6}$) (**Supplementary Table 11**). This ‘notochord development’ gene-set consists in 18 genes, including *EPHA2*, *EFNA1*, and *NOTO*. *EPHA2* encodes the EPH receptor A2, and mutations in this gene are the cause of certain genetically-related cataract disorders, including congenital cataract and age-related cataract¹⁻⁵. *EFNA1* encodes the ephrin A1 which has been implicated in mediating developmental events⁶ and in apoptosis and retinal epithelial development⁷. Interestingly, *EFNA1* is located within the *DPM3-KRTCAP2* cataract-associated locus identified in the current study. The *NOTO* gene encodes a homeobox that is essential for the development of the notochord in zebrafish and mouse models^{8,9}. Finally, the *COL2A1* gene encodes collagen type II alpha 1 chain; mutations in this gene can cause Stickler Syndrome Type 1 which is a heterogeneous group of collagen tissue disorders, characterized by orofacial features, and ophthalmological features such as high myopia, vitreoretinal degeneration, retinal detachment, and presenile cataracts^{10,11}. Future studies could clarify the relationship between genes and pathways commonly involved in notochord development and lens/ataract risk.”

3. The iSyTE resource comprises microarray data and RNA sequencing data. This is very helpful. Can the authors consider RT-PCR validation of perhaps the most insightful findings in independent tissues? Or at least comment? This is because RNA seq data expressed in counts per million may not be very intuitive to other researchers who may like to 'visualize' the expression output.

Thank you for your comments.

As suggested, we have independently validated lens expression of several GWAS-identified candidate genes. We have included a new figure describing this data (**Supplementary Fig. 11**) (discussed on Page 8).

Supplementary Figure 11. RT-PCR based validation of candidate gene expression in the lens. Several GWAS-identified candidates were independently validated by reverse transcriptase (RT)-polymerase chain reaction (PCR) assay for their expression in mouse lens at embryonic day (E)16.5 and postnatal day (P)3.

Further, it has been well established in several earlier publications (ours as well as others in the cataract research community) that genes determined as “lens expressed” by iSyTE are indeed expressed in the lens as validated by independent methods. For example, iSyTE-predicted cataract gene SIPA113 (previously predicted and validated to be expressed in the lens¹²) was subsequently found to be associated to human cataract^{13,14}. Similarly, numerous other cataract/lens defects-linked genes predicted by iSyTE were independently validated to be expressed in the lens – for example, Caprin2 (also identified in the present study), Celf1, Mafg, Rbm24, Tdrd7, among many others^{12,15-20}. We have now cited these references on Page 7 to better orient readers for the relevance of iSyTE-based gene expression in the lens biology and cataract.

Further, as suggested, we have included an explanation of the significance of expression units for microarray and RNA-seq in Methods. For example, our previous findings have shown that fluorescence expression intensity units of around 100 (with significant expression p-value) in the Affymetrix and Illumina microarray platforms has served as good indicators that a candidate gene will be validated by independent assays such as RT-PCR^{21,22}. To further orient readers to visualize the expression of candidate genes, we now also include information on Page 7 as to how further insights on these can be obtained using the iSyTE tool. Briefly, an expression heat-map for candidate genes can be accessed through the iSyTE website (<https://research.bioinformatics.udel.edu/iSyTE>) for both microarray and RNA-seq data, which can be

used for comparative analysis of expression and lens-enriched expression of known cataract-linked genes and those newly predicted in the present study.

4. The authors examined changes in the expression of candidate genes in associated loci in 9 gene perturbation mouse models of lens defects. Are these 9 mouse models relevant given the biological context (human loci, mouse expression); why were these mouse models chosen?

Yes, all the 9 mouse models are relevant to lens biology and cataract based on previous reports, which was the basis of their selection in the present analysis. For example, FOXE3 mutations are linked to cataract and eye defects in human and mouse disease models²³⁻²⁵, HSF4 mutations are linked to cataract in human and mouse disease models^{24,26,27}, PAX6 mutations are linked to eye defects and cataract in human and various animal models^{28,29}, TDRD7 mutations are linked to cataract in human and various animal models^{15,20,30-33}, Sparc knockout mice exhibit cataract³⁴, Klf4 lens-specific conditional knockout mice exhibit cataract³⁵, Mafg:Mafk compound mice exhibit cataract¹⁷, Notch2 lens-specific conditional knockout mice exhibit cataract³⁶, E2f1:E2f2:E2f3 triple lens-specific conditional knockout mice exhibit cataract³⁷ and Brg1 dominant negative expression in the lens results in cataract²⁷. This information is now included on Page 7-8.

I hope you will find the comments helpful

All the best

Khor Chiea Chuen

Reviewer #2 (Remarks to the Author):

This manuscript describes a genome-wide association study for age-related cataract using multiple biobank cohorts. The authors identify 37 novel loci and 17 known loci. Of the 54 loci, 45 (29 novel and 16 known) were replicated in an independent cohort. The authors then stratify by ethnicity and identify an additional 3 loci, and performed conjoint analyses of loci identified in Europeans, revealing 5 more independent signals. Sex-specific analyses identified two more novel loci detected in females only and revealed sex-based differences in effect size at 5 loci from the main metaanalysis. Gene and pathway-based analyses were used to further explore the data, along with interrogation of candidate genes at each locus in lens specific (mouse) expression data. Finally, a PheWAS approach identifies correlations with a range of other phenotypes.

The paper is clearly written, the methods are comprehensive, and the results are clearly displayed.

We thank the reviewer for the positive feedback and constructive review.

Specific questions:

1. Table 1: the loci in grey highlight are previously reported. Can you please indicate where they were previously reported? If they are from the paper “Boutin et al. Insights into the genetic basis of retinal detachment” it should be clearly noted that the UK Biobank cohort was used in both studies and they are not independent replications. Conversely, if there is an independent replication of a previously reported locus, that should be noted too.

We confirmed that the loci highlighted in grey in the Table 1 are from the paper Boutin et al.³⁸ (Insights into the genetic basis of retinal detachment. Hum Mol Genet. 2020). We have added this information in a note below **Table 1**:

“Highlighted in grey are previously reported loci (from Boutin et al. Hum Mol Genet. 2020)”

To fully address the reviewer’s comment, and provide an independent replication of the 20 cataract operation-associated loci previously identified in the Boutin et al study, we also investigated the lead SNPs within 20 loci in the GERA cohort. Those replication results are now reported in the manuscript (as follows) and in a supplementary table (**Supplementary Table 2**):

“We also investigated in GERA the lead SNPs within 20 loci associated with cataract at a genome-wide significance level in a previous study³⁸. Three of the 19 available SNPs that passed QC replicated at a genome-wide level of significance in our GERA multiethnic meta-analysis or GERA non-Hispanic white sample (including *SOX2-OT* rs9842371, 5' LOC338694 rs79721202, and *SLC24A3* rs4814857) (**Supplementary Table 2**). Further, 6 additional SNPs replicated at Bonferroni significance ($P < 0.05/19 = 0.00263$), and 6 showed nominal evidence of association.”

2. Supplementary figure 2 indicates a lambda value of 1.139. This suggests some degree of inflation. How was this combatted and what considerations should be made for this in the final interpretation?

Because of the large sample size, we obtained a genomic inflation factor lambda (λ) of 1.139 for the combined (GERA+UKB) multiethnic meta-analysis, which is reasonable for a dichotomous trait with polygenic inheritance with a sample size this large³⁹. Since λ scales with sample size, some have found it informative to report λ_{1000} ^{40,41}, the inflation factor for an equivalent study of 1000 cases and 1000 controls, which can be calculated by rescaling λ , as below:

$$\lambda_{1000} = 1 + ((\lambda_{obs} - 1) * (1/n_{cases(obs)} + 1/n_{controls(obs)}) / (1/n_{cases(1000)} + 1/n_{controls(1000)}))$$

, where $n_{cases(obs)}$ and $n_{controls(obs)}$ are the study sample size for cases and controls, respectively, and $n_{cases(1000)}$ and $n_{controls(1000)}$ are the target sample size (1000).

So, we have calculated the lambda 1000 for the multiethnic meta-analysis across the 2 cohorts, as follows:

$$\lambda_{1000} = 1 + ((1.139 - 1) * (1/67,844 + 1/517,399) / (1/1000 + 1/1000))$$

$$\lambda_{1000} = 1.0012$$

We obtained the value of 1.0012 for λ_{1000} , which is reasonable for a genomic inflation factor under the assumption of polygenic inheritance.

We have added this information in the Results section, and we have also added some references to justify the initial lambda value of 1.139, as below:

“We first undertook GWAS analysis of cataract in the GERA and UKB cohorts, stratified by ethnic group, followed by a meta-analysis across all analytical strata. In the multiethnic meta-analysis, we identified 54 loci ($P < 5.0 \times 10^{-8}$; $\lambda = 1.139$ and $\lambda_{1000} = 1.0012$, which is reasonable for a sample of this size under the assumption of polygenic inheritance³⁹⁻⁴¹.”

3. The phenotype in GERA and to some extent in UK Biobank is defined as cataract surgery (not cataract diagnosis). This means that there will be people with cataract in the control cohort. Please consider this when discussing the limitations.

While we agree with the reviewer that there are potentially individuals with cataract diagnosis in the GERA control group, we believe that ‘cataract surgery’ represents a deeper phenotype for age-related cataract compared to ‘cataract diagnosis’. Cataract surgery represents a strong validation of the diagnosis as it is conducted at a later stage of the disease. An extension of the cataract phenotypes (e.g. cataract diagnosis) investigated in GWAS is more likely to result in the discovery of additional loci (e.g. specific to earlier stage of the disease) and could provide important biological mechanisms underlying cataract development.

We have added some text in the Discussion to reflect this limitation, as below:

“Our study should be interpreted within the context of its limitations. First, the cataract phenotypes were assessed differently across the 3 study cohorts. While our cataract phenotype in GERA was based on electronic health records (EHRs) data and International Classification of Disease, Ninth (ICD9) or Tenth Revision (ICD10) diagnosis codes, most of the cataract cases in UKB, and all of the cataract cases in 23andMe research cohort (our replication sample) were based on self-reported data. This may result in phenotype misclassification, however, our meta-analysis combining GERA and UKB showed consistency of the SNPs effect estimates between cohorts, and the identified associations were well validated in the 23andMe research cohort. Second, our discovery analysis mainly focused on the cataract surgery phenotype, and as cataracts generally begin to develop in people age 40 years and older, some individuals with early cataract or who will go on to develop cataract later in life might be in the control groups. However, we feel that ‘cataract

surgery' represents a deeper phenotype for age-related cataract and a strong validation of the diagnosis as it is conducted at a later stage of the disease. An extension of the cataract phenotypes (e.g. cataract diagnosis) investigated in GWAS is more likely to result in the discovery of additional loci (e.g. specific to earlier stage of the disease) and could provide important biological mechanisms underlying cataract development. Finally, subtypes of cataract were not available in the 3 study cohorts, which may result in underestimates of the effects of individual SNPs due to phenotype misclassification. Future studies will determine whether the identified loci contribute to different cataract subtypes (i.e. nuclear, cortical, or subcapsular) and the extent to which these loci display shared effects across subtypes.”

4. In all the cohorts, the cases are significantly older than the controls. The controls are not old enough on average to exclude cataract. Was age adjusted for in the GWAS analyses of UKBiobank? (It is mentioned for GERA and 23andMe). Please also consider this age difference in the limitations, particularly in the context of the 23andMe replication cohort where the mean age of controls is only 45 years. Given the size of this cohort, it would be expected to be substantially more powerful than the discovery cohort, however the potential for misclassification is high (given self-report and young age).

We agree with the reviewer that age is an important factor to consider as cataract is age-related and starts developing generally in people age 40 years and older. We confirmed that the GWAS analyses conducted in UK Biobank were adjusted for age (as well as sex and genetic ancestry components). This information is in the Methods, as follows:

“UK Biobank. (...) Following QC, over 10 million variants in 487,622 individuals were tested for cataract adjusting for age, sex, and genetic ancestry principal components.”

We also discussed this point as a limitation, as follows:

“Our study should be interpreted within the context of its limitations. First, the cataract phenotypes were assessed differently across the 3 study cohorts. While our cataract phenotype in GERA was based on electronic health records (EHRs) data and International Classification of Disease, Ninth (ICD9) or Tenth Revision (ICD10) diagnosis codes, most of the cataract cases in UKB, and all of the cataract cases in 23andMe research cohort (our replication sample) were based on self-reported data. This may result in phenotype misclassification, however, our meta-analysis combining GERA and UKB showed consistency of the SNPs effect estimates between cohorts, and the identified associations were well validated in the 23andMe research cohort. Second, our discovery analysis mainly focused on the cataract surgery phenotype, and as cataracts generally begin to develop in people age 40 years and older, some individuals with early cataract or who will go on to develop cataract later in life might be in the control groups. However, we feel that ‘cataract surgery’ represents a deeper phenotype for age-related cataract and a strong validation of the diagnosis as it is conducted at a later stage of the disease. An extension of the cataract phenotypes (e.g. cataract diagnosis) investigated in GWAS is more likely to result in the discovery of additional loci (e.g. specific to earlier stage of the disease) and could provide important biological mechanisms underlying cataract development. Finally, subtypes of cataract were not available in the 3 study cohorts, which may result in underestimates of the effects of individual SNPs due to phenotype misclassification. Future studies will determine whether the identified loci contribute to different cataract subtypes (i.e. nuclear, cortical, or subcapsular) and the extent to which these loci display shared effects across subtypes.”

5. How were candidate genes at each locus selected for the subsequent iSyTE analyses?

The iSyTE database was used to analyze mouse orthologs of the human candidate genes in the 54 loci linked to cataract in the multiethnic GWAS meta-analysis combining GERA and UK Biobank.

A candidate gene was defined as the nearest gene with respect to the original top variant for each identified genomic region. We have added this information in the results and the methods sections.

6. Please expand on the significance of the PheWAS results. Is there likely to be a biological link to cataract, or are there likely to be bias or confounding factors in phenotyping that lead to these correlations?

This is an excellent suggestion. We have now expanded on the significance of the PheWAS results as follows:

“A phenome-wide association study (PheWAS) analysis of 43 cataract-associated SNPs, available in the GeneAtlas was run across 776 traits measured and previously analyzed in the UKB. Twenty-three of the most significantly associated cataract-associated variants were significantly associated ($P < 5.0 \times 10^{-8}$) with other traits (Fig. 4). Most were associated with disorders of the lens, with the strongest association observed for the intronic variant rs4814857 at *SLC24A3* ($P = 2.48 \times 10^{-39}$) (Supplementary Table 13). *SLC24A3* encodes the carrier family 24 member 3 and has been thought to be involved in retinal diseases. Variants at *PLCE1* and *HMGA2* were significantly associated with hypertension, diabetes, and anthropometric traits, such as fat mass and waist circumference. Although the relationship between age-related cataract and metabolic syndrome has been well established⁴²⁻⁴⁵, the molecular mechanisms underlying these clinical observations remain poorly understood. Our PheWAS findings revealed that *PLCE1* and *HMGA2* could be the genetic links between age-related cataract and metabolic syndrome. Our PheWAS analysis also highlighted that variants at *OCA2* and *NPLOC4* were significantly associated with pigmentation phenotypes. The *OCA2* gene encodes the melanosomal transmembrane protein, whose variants determine iris color and have been linked to corneal and refractive astigmatism, syndromic forms of myopia, refractive error, and type 2 oculocutaneous albinism (Supplementary Table 14). *NPLOC4* encodes the homolog, ubiquitin recognition factor and has been previously associated with macular thickness and the risk of strabismus and corneal and refractive astigmatism. Despite compelling evidence, our PheWAS results raise the need of further studies to keep unravelling these complex human genome-phenome relationships and unveiling the molecular mechanisms that support them⁴⁶.”

REFERENCES

1. Iyengar, S.K. *et al.* Identification of a major locus for age-related cortical cataract on chromosome 6p12-q12 in the Beaver Dam Eye Study. *Proc Natl Acad Sci U S A* **101**, 14485-90 (2004).
2. Shiels, A. *et al.* The EPHA2 gene is associated with cataracts linked to chromosome 1p. *Mol Vis* **14**, 2042-55 (2008).
3. Jun, G. *et al.* EPHA2 is associated with age-related cortical cataract in mice and humans. *PLoS Genet* **5**, e1000584 (2009).
4. Tan, W. *et al.* Association of EPHA2 polymorphisms and age-related cortical cataract in a Han Chinese population. *Mol Vis* **17**, 1553-8 (2011).
5. Astiazaran, M.C., Garcia-Montano, L.A., Sanchez-Moreno, F., Matiz-Moreno, H. & Zenteno, J.C. Next generation sequencing-based molecular diagnosis in familial congenital cataract expands the mutational spectrum in known congenital cataract genes. *Am J Med Genet A* **176**, 2637-2645 (2018).
6. Duffy, S.L., Steiner, K.A., Tam, P.P. & Boyd, A.W. Expression analysis of the Epha1 receptor tyrosine kinase and its high-affinity ligands Efn1 and Efn3 during early mouse development. *Gene Expr Patterns* **6**, 719-23 (2006).
7. Korthagen, N.M. *et al.* Retinal pigment epithelial cells display specific transcriptional responses upon TNF- α stimulation. *Br J Ophthalmol* **99**, 700-4 (2015).
8. Talbot, W.S. *et al.* A homeobox gene essential for zebrafish notochord development. *Nature* **378**, 150-7 (1995).
9. Beckers, A., Alten, L., Viebahn, C., Andre, P. & Gossler, A. The mouse homeobox gene *Noto* regulates node morphogenesis, notochordal ciliogenesis, and left right patterning. *Proc Natl Acad Sci U S A* **104**, 15765-70 (2007).
10. Higuchi, Y., Hasegawa, K., Yamashita, M., Tanaka, H. & Tsukahara, H. A novel mutation in the COL2A1 gene in a patient with Stickler syndrome type 1: a case report and review of the literature. *J Med Case Rep* **11**, 237 (2017).
11. Goyal, M., Kapoor, S., Ikegawa, S. & Nishimura, G. Stickler Syndrome Type 1 with Short Stature and Atypical Ocular Manifestations. *Case Rep Pediatr* **2016**, 3198597 (2016).
12. Lachke, S.A. *et al.* iSyTE: integrated Systems Tool for Eye gene discovery. *Invest Ophthalmol Vis Sci* **53**, 1617-27 (2012).
13. Evers, C. *et al.* SIPA1L3 identified by linkage analysis and whole-exome sequencing as a novel gene for autosomal recessive congenital cataract. *Eur J Hum Genet* **23**, 1627-33 (2015).
14. Greenlees, R. *et al.* Mutations in SIPA1L3 cause eye defects through disruption of cell polarity and cytoskeleton organization. *Hum Mol Genet* **24**, 5789-804 (2015).
15. Lachke, S.A. *et al.* Mutations in the RNA granule component TDRD7 cause cataract and glaucoma. *Science* **331**, 1571-6 (2011).
16. Dash, S., Dang, C.A., Beebe, D.C. & Lachke, S.A. Deficiency of the RNA binding protein caprin2 causes lens defects and features of Peters anomaly. *Dev Dyn* **244**, 1313-27 (2015).

17. Agrawal, S.A. *et al.* Compound mouse mutants of bZIP transcription factors Mafg and Mafk reveal a regulatory network of non-crystallin genes associated with cataract. *Hum Genet* **134**, 717-35 (2015).
18. Siddam, A.D. *et al.* The RNA-binding protein Celf1 post-transcriptionally regulates p27Kip1 and Dnase2b to control fiber cell nuclear degradation in lens development. *PLoS Genet* **14**, e1007278 (2018).
19. Dash, S. *et al.* The master transcription factor SOX2, mutated in anophthalmia/microphthalmia, is post-transcriptionally regulated by the conserved RNA-binding protein RBM24 in vertebrate eye development. *Hum Mol Genet* **29**, 591-604 (2020).
20. Barnum, C.E. *et al.* The Tudor-domain protein TDRD7, mutated in congenital cataract, controls the heat shock protein HSPB1 (HSP27) and lens fiber cell morphology. *Hum Mol Genet* **29**, 2076-2097 (2020).
21. Terrell, A.M. *et al.* Molecular characterization of mouse lens epithelial cell lines and their suitability to study RNA granules and cataract associated genes. *Exp Eye Res* **131**, 42-55 (2015).
22. Weatherbee, B.A.T., Barton, J.R., Siddam, A.D., Anand, D. & Lachke, S.A. Molecular characterization of the human lens epithelium-derived cell line SRA01/04. *Exp Eye Res* **188**, 107787 (2019).
23. Landgren, H., Blixt, A. & Carlsson, P. Persistent FoxE3 expression blocks cytoskeletal remodeling and organelle degradation during lens fiber differentiation. *Invest Ophthalmol Vis Sci* **49**, 4269-77 (2008).
24. Anand, D., Agrawal, S.A., Slavotinek, A. & Lachke, S.A. Mutation update of transcription factor genes FOXE3, HSF4, MAF, and PITX3 causing cataracts and other developmental ocular defects. *Hum Mutat* **39**, 471-494 (2018).
25. Krall, M. *et al.* A zebrafish model of foxe3 deficiency demonstrates lens and eye defects with dysregulation of key genes involved in cataract formation in humans. *Hum Genet* **137**, 315-328 (2018).
26. Bu, L. *et al.* Mutant DNA-binding domain of HSF4 is associated with autosomal dominant lamellar and Marner cataract. *Nat Genet* **31**, 276-8 (2002).
27. He, S. *et al.* Chromatin remodeling enzyme Brg1 is required for mouse lens fiber cell terminal differentiation and its denucleation. *Epigenetics Chromatin* **3**, 21 (2010).
28. Glaser, T. *et al.* PAX6 gene dosage effect in a family with congenital cataracts, aniridia, anophthalmia and central nervous system defects. *Nat Genet* **7**, 463-71 (1994).
29. Ashery-Padan, R., Marquardt, T., Zhou, X. & Gruss, P. Pax6 activity in the lens primordium is required for lens formation and for correct placement of a single retina in the eye. *Genes Dev* **14**, 2701-11 (2000).
30. Zheng, C. *et al.* RNA granule component TDRD7 gene polymorphisms in a Han Chinese population with age-related cataract. *J Int Med Res* **42**, 153-63 (2014).
31. Chen, J. *et al.* Molecular Genetic Analysis of Pakistani Families With Autosomal Recessive Congenital Cataracts by Homozygosity Screening. *Invest Ophthalmol Vis Sci* **58**, 2207-2217 (2017).
32. Tan, Y.Q. *et al.* Loss-of-function mutations in TDRD7 lead to a rare novel syndrome combining congenital cataract and nonobstructive azoospermia in humans. *Genet Med* **21**, 1209-1217 (2019).
33. Kandaswamy, D.K. *et al.* Application of WES Towards Molecular Investigation of Congenital Cataracts: Identification of Novel Alleles and Genes in a Hospital-Based Cohort of South India. *Int J Mol Sci* **21**(2020).
34. Greiling, T.M., Stone, B. & Clark, J.I. Absence of SPARC leads to impaired lens circulation. *Exp Eye Res* **89**, 416-25 (2009).
35. Gupta, D., Harvey, S.A., Kenchegowda, D., Swamynathan, S. & Swamynathan, S.K. Regulation of mouse lens maturation and gene expression by Kruppel-like factor 4. *Exp Eye Res* **116**, 205-18 (2013).
36. Saravanamuthu, S.S. *et al.* Conditional ablation of the Notch2 receptor in the ocular lens. *Dev Biol* **362**, 219-29 (2012).
37. Wenzel, P.L. *et al.* Cell proliferation in the absence of E2F1-3. *Dev Biol* **351**, 35-45 (2011).
38. Boutin, T.S. *et al.* Insights into the genetic basis of retinal detachment. *Hum Mol Genet* **29**, 689-702 (2020).
39. Yang, J. *et al.* Genomic inflation factors under polygenic inheritance. *Eur J Hum Genet* **19**, 807-12 (2011).
40. de Bakker, P.I. *et al.* Practical aspects of imputation-driven meta-analysis of genome-wide association studies. *Hum Mol Genet* **17**, R122-8 (2008).
41. Freedman, M.L. *et al.* Assessing the impact of population stratification on genetic association studies. *Nat Genet* **36**, 388-93 (2004).
42. Mehra, S., Kapur, S., Mittal, S. & Sehgal, P.K. Common genetic link between metabolic syndrome components and senile cataract. *Free Radic Res* **46**, 133-40 (2012).
43. Cheung, N. & Wong, T.Y. Obesity and eye diseases. *Surv Ophthalmol* **52**, 180-95 (2007).
44. Davison, J.E. Eye involvement in inherited metabolic disorders. *Ther Adv Ophthalmol* **12**, 2515841420979109 (2020).

45. Lima-Fontes, M., Barata, P., Falcao, M. & Carneiro, A. Ocular findings in metabolic syndrome: a review. *Porto Biomed J* **5**, e104 (2020).
46. Bush, W.S., Oetjens, M.T. & Crawford, D.C. Unravelling the human genome-phenome relationship using phenome-wide association studies. *Nat Rev Genet* **17**, 129-45 (2016).

Reviewers' Comments:

Reviewer #1:

Remarks to the Author:

Drs Choquet, Lachke, Jorgenson, and colleagues submit a vigorously reanalyzed and thorough rebuttal for all the questions i presented for discussion in the previous round of review.

Specifically, Supplementary Table 6 is a very transparent display of the sex-differentiated association analysis that teaches the reader about a) women specific loci, also replicating in men.

b) women specific loci, not replicating in men, and not replicating in further collections (23andMe)

c) true positive women specific loci that are not seen in men, and replicating again only in women in 23andME and not in men (CASP7). The clarity of the heterogeneity analysis is insightful.

Reviewer #2:

Remarks to the Author:

All my questions have been well addressed.

REVIEWERS' COMMENTS

Reviewer #1 (Remarks to the Author):

Drs Choquet, Lachke, Jorgenson, and colleagues submit a vigorously reanalyzed and thorough rebuttal for all the questions i presented for discussion in the previous round of review.

Specifically, Supplementary Table 6 is a very transparent display of the sex-differentiated association analysis that teaches the reader about a) women specific loci, also replicating in men.

b) women specific loci, not replicating in men, and not replicating in further collections (23andMe)

c) true positive women specific loci that are not seen in men, and replicating again only in women in 23andME and not in men (CASP7). The clarity of the heterogeneity analysis is insightful.

Reviewer #2 (Remarks to the Author):

All my questions have been well addressed.

We thank the reviewers for their positive feedback on the revised manuscript.